# More efficient North Atlantic carbon pump during the Last Glacial Maximum

J. Yu [1,2], L. Menviel [3], Z.D. Jin[2,4], D.J.R. Thornalley [5], G.L. Foster[6], E.J. Rohling [1,6], I.N. McCave [7], J.F. McManus[8], Y. Dai[1], H. Ren [9], F. He [10,11], F. Zhang [2,12], P.J. Chen[13] & A.P. Roberts[1]

During the Last Glacial Maximum (LGM; ~20,000 years ago), the global ocean sequestered a large amount of carbon lost from the atmosphere and terrestrial biosphere. Suppressed $CO_2$ outgassing from the Southern Ocean is the prevailing explanation for this carbon sequestration. By contrast, the North Atlantic Ocean—a major conduit for atmospheric $CO_2$ transport to the ocean interior via the overturning circulation—has received much less attention. Here we demonstrate that North Atlantic carbon pump efficiency during the LGM was almost doubled relative to the Holocene. This is based on a novel proxy approach to estimate air–sea $CO_2$ exchange signals using combined carbonate ion and nutrient reconstructions for multiple sediment cores from the North Atlantic. Our data indicate that in tandem with Southern Ocean processes, enhanced North Atlantic $CO_2$ absorption contributed to lowering ice-age atmospheric $CO_2$.

[1] Research School of Earth Sciences, The Australian National University, Canberra, ACT 2601, Australia. [2] SKLLQG, Institute of Earth Environment, Chinese Academy of Sciences, Xi'an 710061, China. [3] Climate Change Research Centre, University of New South Wales, Sydney, NSW 2052, Australia. [4] Open Studio for Oceanic-Continental Climate and Environment Changes, Qingdao National Laboratory for Marine Science and Technology, Qingdao 266061, China. [5] Department of Geography, University College London, London WC1E 6BT, UK. [6] Ocean and Earth Science, University of Southampton, National Oceanography Centre, Southampton SO14 3ZH, UK. [7] Department of Earth Sciences, University of Cambridge, Cambridge CB2 3EQ, UK. [8] Lamont-Doherty Earth Observatory of Columbia University, 61 Route 9W/PO Box 1000, Palisades, NY 10964-8000, USA. [9] Department of Geosciences, National Taiwan University, Taipei, Taiwan. [10] Center for Climatic Research, Nelson Institute for Environmental Studies, University of Wisconsin-Madison, Madison, WI 53706, USA. [11] College of Earth, Ocean, and Atmospheric Sciences, Oregon State University, Corvallis, OR 97331, USA. [12] CAS Center for Excellence in Quaternary Science and Global Change, Xi'an 710061, China. [13] State Key Laboratory of Marine Geology, Tongji University, Shanghai 200092, China. Correspondence and requests for materials should be addressed to J.Y. (email: jimin.yu@anu.edu.au)

The North Atlantic Ocean (>~35°N, including the Nordic Seas and Arctic Ocean) is a major atmospheric $CO_2$ sink, which has been mitigating anthropogenic atmospheric $CO_2$ increases[1]. Preindustrial North Atlantic surface water partial pressure of $CO_2$ ($pCO_2$) was up to ~100 µatm lower than the contemporary atmospheric $pCO_2$ of ~280 µatm, which caused substantial atmospheric $CO_2$ invasion[2,3]. Despite its modest area, the North Atlantic Ocean accounts for at least ~30% of the global ocean $CO_2$ uptake today and during preindustrial times[1,4]. Over longer timescales, large-scale oceanic carbon sequestration also occurred during Plio-Pleistocene glaciations[5–7]. This is commonly attributed to reduced glacial Southern Ocean $CO_2$ outgassing[6,8,9], while even the sign of past North Atlantic $CO_2$ uptake efficiency changes remains unconstrained. Here, we present a novel proxy approach to trace atmospheric $CO_2$ invasion in the North Atlantic and thereby evaluate its role in carbon sequestration in ice-age oceans. We find that the last glacial North Atlantic carbon absorption became more efficient, highlighting a critical role of the North Atlantic Ocean in regulating glacial–interglacial atmospheric $CO_2$ changes.

## Results

**Air–sea $CO_2$ exchange tracers.** Any effect of ocean processes on atmospheric $pCO_2$ must occur via air–sea $CO_2$ exchange. In the North Atlantic, high-nutrient utilization decreases surface-water dissolved inorganic carbon (DIC) and causes surface-water $pCO_2$ to be lower than atmospheric $pCO_2$ (Supplementary Fig. 1). This leads to net air-to-sea $CO_2$ transfer, creating an air–sea exchange signature of DIC ($DIC_{as}$). $DIC_{as}$ signals can be distinguished by accounting for within-ocean DIC redistributions that are heavily mediated by biology (Fig. 1). Biological cycling of organic matter depletes DIC and nutrients such as phosphate ($PO_4$) in surface waters and enriches them at depth. Seawater mixing also affects DIC and $PO_4$ concentrations in the ocean. Nevertheless, $PO_4$ variations are ultimately determined by biological processes: without biology, $PO_4$ should be the same everywhere in the ocean regardless of ocean circulation (ignoring the small effect from salinity change). Because marine biology incorporates and releases $PO_4$ and DIC in a relatively fixed proportion following Redfield stoichiometry[3,10] and because $PO_4$ is not affected by air–sea exchange, $PO_4$ can be used to estimate biology-driven within-ocean DIC redistributions (Fig. 1). Any within-ocean DIC redistribution associated with $CaCO_3$ cycling can be accounted for using alkalinity (ALK) and nitrate.

Following the established method[3] to account for within-ocean DIC redistributions by soft-tissue and $CaCO_3$ cycling, we calculate preindustrial Atlantic $DIC_{as}$ using the GLODAP dataset[2] (Fig. 2a). See Methods for details to calculate $DIC_{as}$. More positive $DIC_{as}$ values indicate a greater degree of atmospheric $CO_2$ invasion. At basin-scale, the preindustrial $DIC_{as}$ of North Atlantic deep water (NADW) is ~50–80 µmol/kg higher than for Antarctic bottom water (AABW) and Antarctic intermediate water (AAIW). This difference reflects North Atlantic $CO_2$ uptake and Southern Ocean release[3,11]. North Atlantic $CO_2$ absorption is driven by (i) an efficient solubility pump due to strong cooling of northward-flowing Gulf Stream waters and (ii) a strong biological pump associated with high nutrient utilization[12–14]. NADW thus represents an efficient pathway for atmospheric $CO_2$ sequestration[6,15]. Through global deep ocean circulation, $CO_2$ absorbed in the North Atlantic is transported throughout the world ocean[1,3], with profound implications for the global carbon cycle.

No proxy exists to reconstruct past seawater DIC and ALK at acceptable precision for direct application, so we employ a linked carbonate system parameter for palaeoceanographic studies. Everything else being equal, atmospheric $CO_2$ invasion would decrease seawater carbonate ion concentration ($[CO_3^{2-}]$), because $CO_2$ reacts with carbonate ion to form bicarbonate[16]. We thus develop a new tracer, $[CO_3^{2-}]_{as}$, which essentially reflects seawater $[CO_3^{2-}]$ contrasts for the same biological (i.e., $PO_4$) and physical (i.e., temperature–salinity–pressure; T–S–P) conditions (Fig. 2b; see Methods for calculation details). To extract air–sea exchange signals, it is necessary to compare $[CO_3^{2-}]$ at the same $PO_4$–T–S–P conditions because we must first remove influences on $[CO_3^{2-}]$ from (i) within-ocean DIC and ALK redistributions by biology and (ii) T–S–P variations via their effects on $CO_2$ system dissociation constants[16]. In the preindustrial Atlantic, the strong negative correlation between $[CO_3^{2-}]_{as}$ and $DIC_{as}$ (Fig. 2, Supplementary Fig. 2) indicates that $[CO_3^{2-}]_{as}$ variations are affected only by $DIC_{as}$, and thus are ultimately linked to air–sea $CO_2$ exchange.

The Gulf Stream is a major NADW source[17]; thus, comparing the $[CO_3^{2-}]_{as}$ gradient between the Gulf Stream and NADW can provide a measure of $CO_2$ sequestration intensity during transformation of Gulf Stream waters into NADW. Because Gulf Stream waters are more or less in equilibrium with atmospheric $pCO_2$ from ~10°N to 35°N[1,2], the Gulf Stream−NADW $[CO_3^{2-}]_{as}$ gradient mainly reflects North Atlantic (>~35°N) air–sea $CO_2$

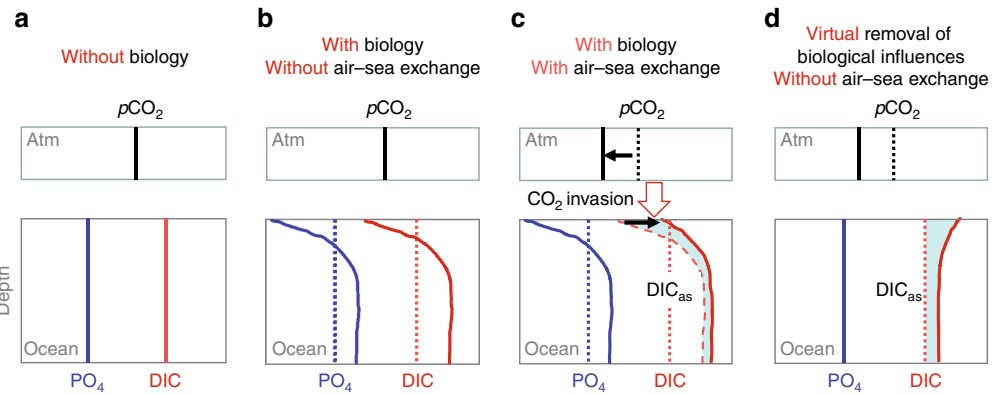

**Fig. 1** Concepts to distinguish $DIC_{as}$. For simplicity, only $CO_2$ invasion associated with organic matter cycling is considered. In the ocean box, vertical solid and dashed lines (**a–d**) represent mean $PO_4$ (blue) and DIC (red) in an abiotic ocean (**a**). Biology redistributes DIC and $PO_4$ following Redfield stoichiometry (curves; **b**). This decreases surface-ocean DIC and $pCO_2$, and hence causes air-to-sea $CO_2$ transfer (**c**). Through mixing and ocean circulation, $CO_2$ invasion raises water-column DIC, i.e., shifting dashed curve (equals the red-solid curve in **b**) to red-solid curve (**c**). The shaded region in **c** represents air–sea exchange $DIC_{as}$ signatures. After removing carbon redistribution by biology based on $PO_4$-related curvature of the profiles (**b**), $DIC_{as}$ can be revealed by the shaded region in **d**

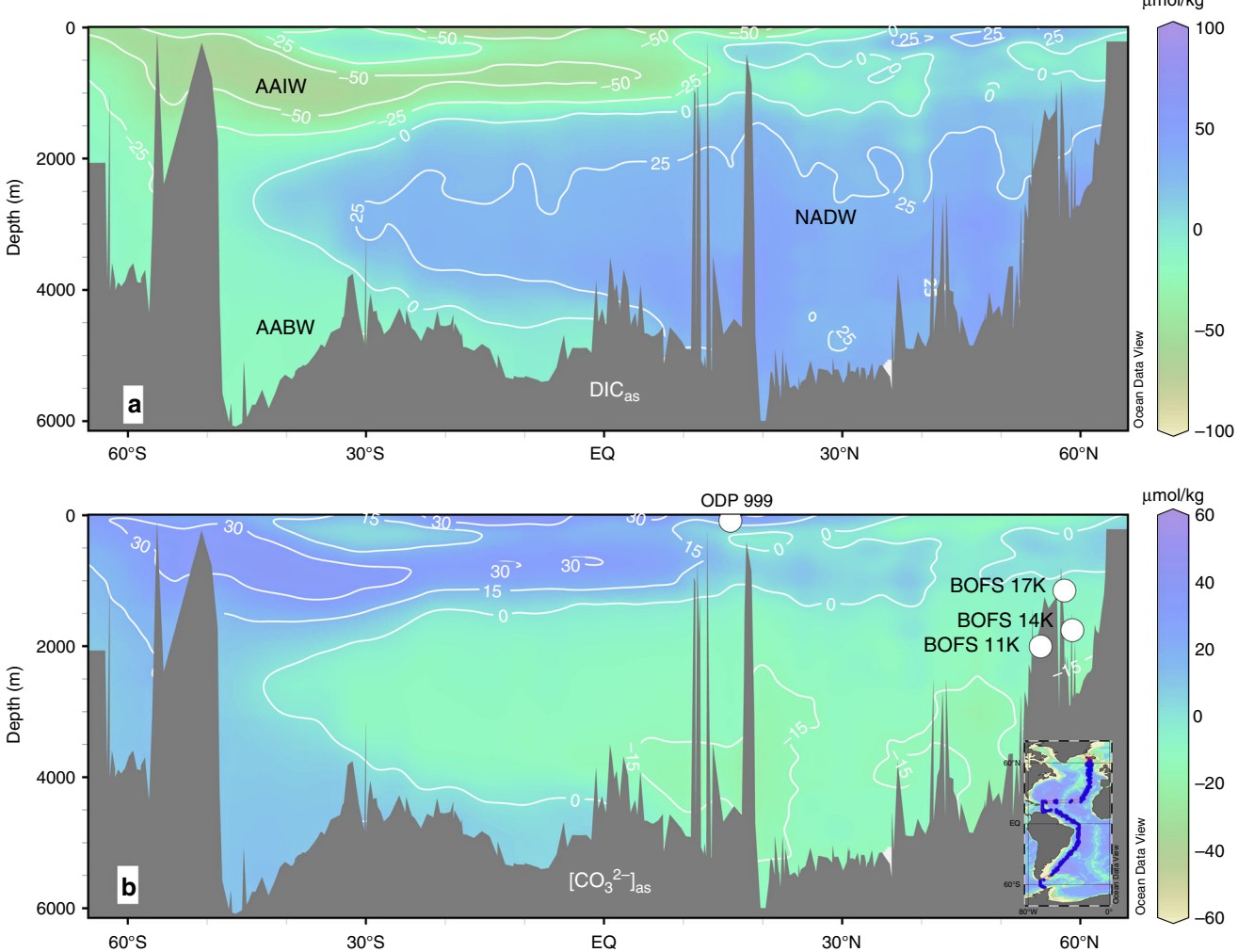

**Fig. 2** Preindustrial Atlantic air–sea exchange tracers. **a** DIC$_{as}$. **b** [CO$_3^{2-}$]$_{as}$. Circles represent studied sediment cores. Inset: GLODAP hydrographic data[2] used to generate the sections[96]. NADW North Atlantic deep water, AABW Antarctic bottom water, AAIW Antarctic intermediate water. See Methods for calculation details

exchange (Supplementary Fig. 3). Physical oceanographers have shown that the path of Gulf Stream waters, rather than being a direct conveyor to the polar North Atlantic, is instead a "corkscrew", where Gulf Stream waters are recirculated south in the subtropical gyre and subduct after being made more dense by air–sea heat loss (e.g., refs. [18,19]). However, our interest lies in net CO$_2$ uptake by the North Atlantic region, and variations in spatial pathways from Gulf Stream to NADW formation sites[18,19] should not significantly complicate our conclusion. The greater the [CO$_3^{2-}$]$_{as}$ gradient between Gulf Stream and NADW (instead of their absolute [CO$_3^{2-}$]$_{as}$ values), the more efficient air–sea CO$_2$ absorption by the North Atlantic. Linked to large-scale overturning circulation, Gulf Stream−NADW [CO$_3^{2-}$]$_{as}$ gradient changes regulate long-term CO$_2$ sequestration into the deep ocean.

**Downcore reconstructions**. Next, we reconstruct past Gulf Stream–NADW [CO$_3^{2-}$]$_{as}$ gradients to investigate North Atlantic carbon pump efficiency during the LGM (18–27 ka). Previous work suggests that most of North Atlantic subtropical gyre water circulates through the Caribbean Sea before being transported to the subpolar North Atlantic via the Gulf Stream[20]. We, therefore, use Caribbean Sea ODP Site 999 (12.8°N, 78.7°W) to

constrain past Gulf Stream physicochemical conditions (Fig. 3, Supplementary Figs. 4 and 5). The feasibility of using ODP Site 999 to reflect the first-order Gulf Stream carbonate chemistry changes between the Holocene and LGM is supported by observations that (i) Caribbean surface waters have similar [CO$_3^{2-}$]$_{as}$ values to hydrographic sites located within Gulf Stream during the preindustrial (Supplementary Fig. 3), and (ii) cores from the broader western subtropical Atlantic show comparable Holocene and LGM [CO$_3^{2-}$]$_{as}$ signatures as those from ODP 999 (Supplementary Fig. 6). Surface-water T and S are estimated from *Globigerinoides ruber* Mg/Ca and sea level fluctuations, respectively[21,22]. Previously published *G. ruber* δ$^{11}$B (ref. [21]) is used to calculate surface-water pH, while ALK is estimated from S using the modern relationship between S and ALK[21,22]. Along with T, S, and ALK estimates, pH is then used to calculate surface-water [CO$_3^{2-}$] and DIC. Given the constraint from pH, seawater ALK and DIC must vary systematically within the ocean carbonate system (Supplementary Fig. 5). This allows precise estimation of [CO$_3^{2-}$], because even large ALK uncertainties (100 μmol/kg; ± 2σ, used throughout) only have a minor effect on [CO$_3^{2-}$] (~14 μmol/kg). Given its oligotrophic setting, past surface-water PO$_4$ at ODP 999 is assumed to be zero[2,21,22].

Three cores are used to reconstruct deep-water conditions of northern-sourced waters (Fig. 3). BOFS 17 K (58°N, 16.5°W,

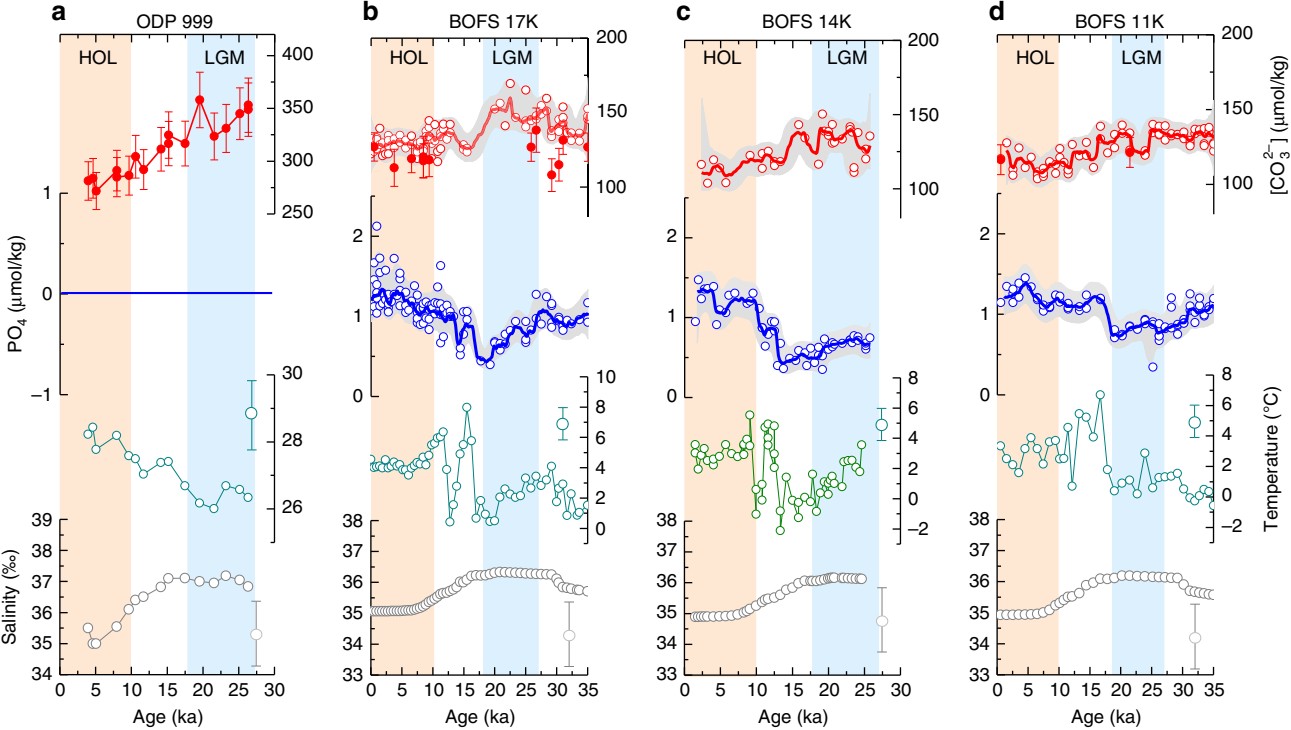

**Fig. 3** Down core reconstructions. **a** ODP 999. **b** BOFS 17 K. **c** BOFS 14 K. **d** BOFS 11 K. Seawater [$CO_3^{2-}$] values are derived from benthic B/Ca (empty circles) and $\delta^{11}$B (solid circles). Light gray envelopes and error bars: 2$\sigma$. Note different $y$-scales for surface- (ODP 999) and deep-water (BOFS cores) reconstructions. See Methods for reconstruction details

1150 m) and BOFS 14 K (58.6°N, 19.4°W, 1756 m) are located close to the previously surmised center of Glacial North Atlantic intermediate water (GNAIW)[23], while BOFS 11 K (55.2°N, 20.4°W, 2004 m) is thought to be affected by glacial Nordic Sea overflows[24]. We employ benthic foraminiferal $\delta^{11}$B and B/Ca to reconstruct deep-water [$CO_3^{2-}$] with an uncertainty of ~10 µmol/kg[25]. $\delta^{11}$B and B/Ca give consistent downcore [$CO_3^{2-}$] reconstructions. Benthic Cd/Ca is used to estimate deep-water Cd and PO$_4$ based on an established approach (Supplementary Fig. 7)[26,27]. Past deep-water T and S changes are estimated from foraminiferal $\delta^{18}$O and sea level fluctuations; use of other methods negligibly affects our conclusion. In total, we present 180 new measurements for benthic foraminiferal $\delta^{11}$B, B/Ca, and Cd/Ca. Details of core materials, methods, new and compiled data, and fully propagated uncertainties are given in Methods and Supplementary Data 1–9.

**A pragmatic recipe to estimate [$CO_3^{2-}$]$_{as}$ change.** Surface-water [$CO_3^{2-}$] at ODP 999 is ~150 µmol/kg higher than deep-water values at BOFS cores (Fig. 3), but this [$CO_3^{2-}$] contrast includes influences from physical (via dissociation constants) and biological (via within-ocean DIC and ALK redistributions) changes in addition to any air–sea $CO_2$ changes between surface and deep waters. Below, we present a pragmatic recipe to estimate [$CO_3^{2-}$]$_{as}$ gradients between water masses. We take advantage of well-defined sensitivities of [$CO_3^{2-}$] to T–S–P (Fig. 4) to calculate normalized seawater [$CO_3^{2-}$] ([$CO_3^{2-}$]$_{Norm}$) at conditions of $T = 3$ °C, $S = 35$‰, and $P = 2500$ dbar (Methods). Any variation in T–S–P would affect seawater [$CO_3^{2-}$] via (i) changing $CO_2$ system dissociation constants, and (ii) altering the solubility pump and thereby air–sea exchange component $CO_2$ concentrations in seawater. Calculation of [$CO_3^{2-}$]$_{Norm}$ only corrects for influences from (i), without affecting any air–sea $CO_2$ signal.

After normalization to constant T–S–P conditions and assuming no net air–sea exchange, biological activity drives changes in both [$CO_3^{2-}$]$_{Norm}$ and PO$_4$ along the biological trend (green curves in Fig. 5; Methods). Note that along a certain biological trend, seawater [$CO_3^{2-}$]$_{Norm}$ and PO$_4$ are only affected by within-ocean DIC and ALK redistributions (Fig. 1b). A net air–sea $CO_2$ change would cause changes in [$CO_3^{2-}$]$_{Norm}$ and PO$_4$ across biological curves. At the same PO$_4$, [$CO_3^{2-}$]$_{Norm}$ contrasts reflect [$CO_3^{2-}$]$_{as}$ gradients due to air–sea $CO_2$ exchange between water masses.

A plot of [$CO_3^{2-}$]$_{Norm}$ vs. PO$_4$ greatly facilitates investigation of air–sea $CO_2$ exchange from combined [$CO_3^{2-}$] and PO$_4$ measurements/reconstructions. Compared to the biological trend, preindustrial North Atlantic surface waters have a steeper trend (Fig. 5a), which reflects $CO_2$ absorption during northward transport. Deep-water data lie on a shallower trend, consistent with mixing between low-[$CO_3^{2-}$]$_{as}$ (high DIC$_{as}$) NADW and high-[$CO_3^{2-}$]$_{as}$ (low DIC$_{as}$) AABW in the deep Atlantic (Fig. 2).

For our downcore reconstructions, benthic Cd/Ca suggests that deep-waters at the BOFS sites had PO$_4$ values of ~1.2 and ~0.8 µmol/kg during the Holocene and LGM, respectively (Fig. 5b; Supplementary Fig. 8). Assuming no air–sea $CO_2$ exchange, [$CO_3^{2-}$]$_{Norm}$ of ODP 999 surface waters at elevated PO$_4$ due to biological processes can be estimated straightforwardly using the H→H′ and G→G′ trajectories in Fig. 5b for the Holocene and LGM, respectively. For the Holocene, ODP 999 [$CO_3^{2-}$]$_{Norm}$ is ~56 ± 8 µmol/kg higher than [$CO_3^{2-}$]$_{Norm}$ of BOFS cores at PO$_4$ = 1.2 µmol/kg. For the LGM, ODP 999 [$CO_3^{2-}$]$_{Norm}$ is ~114 ± 9 µmol/kg higher than [$CO_3^{2-}$]$_{Norm}$ of BOFS cores at PO$_4$ = 0.8 µmol/kg. This suggests a Holocene-to-LGM increase of ~58 ± 12 µmol/kg in the ODP 999−BOFS [$CO_3^{2-}$]$_{as}$ gradient.

We also present a second approach to calculate [$CO_3^{2-}$]$_{as}$ gradients, which involves frequent use of the $CO_2$sys program[28] and intermediate-step ALK and DIC parameters (Supplementary Note 1; Supplementary Figs. 9 and 10). The approach gives

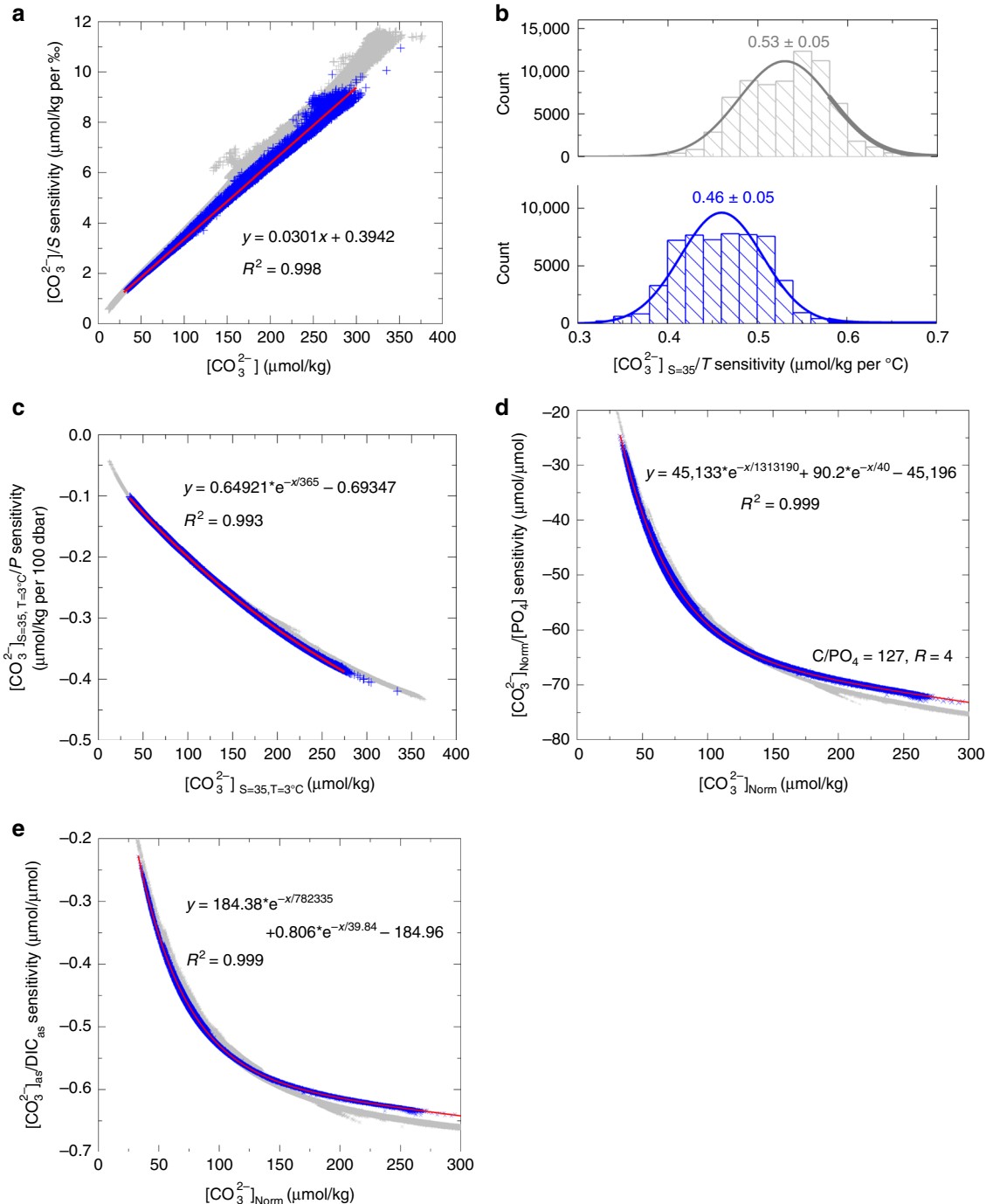

**Fig. 4** Carbonate system sensitivities to various changes. **a** Salinity effect. **b** Temperature effect. **c** Pressure effect. **d** Biological effect. **e** Air–sea $CO_2$ exchange effect. Calculations are based on GLODAP[2] ($n = 55,399$; blue) and a LGM output from LOVECLIM[58] ($n = 71,768$; gray). For **a**–**d**, calculations assume no net air–sea $CO_2$ change. Best fits of data are shown by red curves. See Methods for calculation details

similar results as the above pragmatic recipe, because both methods are essentially based on the same principle, which is to compare $[CO_3^{2-}]$ of water masses at the same physical and biological conditions.

**Enhanced $CO_2$ uptake in the glacial North Atlantic.** What caused the greater ODP 999–BOFS $[CO_3^{2-}]_{as}$ gradient during the LGM? We consider influences from biogenic matter composition variations, surface-water ALK and $PO_4$ changes, ocean circulation changes, and North Atlantic air–sea exchange. In Fig. 5, we have used a soft-tissue Redfield C/$PO_4$ of 127 and a rain

ratio ($R$, = $C_{organic}$:$C_{CaCO_3}$) of 4 (refs. [3,10,29,30]) to predict the biological trend. Raising LGM C/$PO_4$ to 140 (the high end value in today's North Atlantic[30]) and $R$ to 8 (doubling of the modern value) could lower the LGM $[CO_3^{2-}]_{as}$ gradient by ~16 μmol/kg (Supplementary Fig. 15), still leaving ~42 μmol/kg $[CO_3^{2-}]_{as}$ gradient increase to be explained by other processes. Evidence for such large biological changes is lacking. Importantly, any increase in C/$PO_4$ and $R$ would implicitly sequester more atmospheric $CO_2$ via an enhanced soft-tissue pump and weakened carbonate pump[15]. Inclusion of a whole ocean ALK inventory change[6] or any increased glacial surface-water $PO_4$ at ODP 999

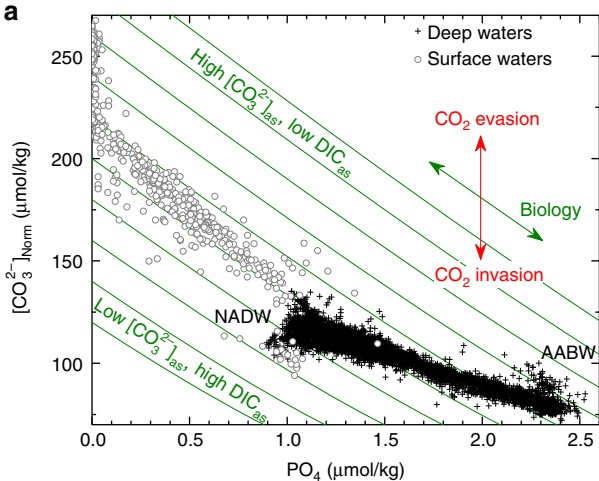

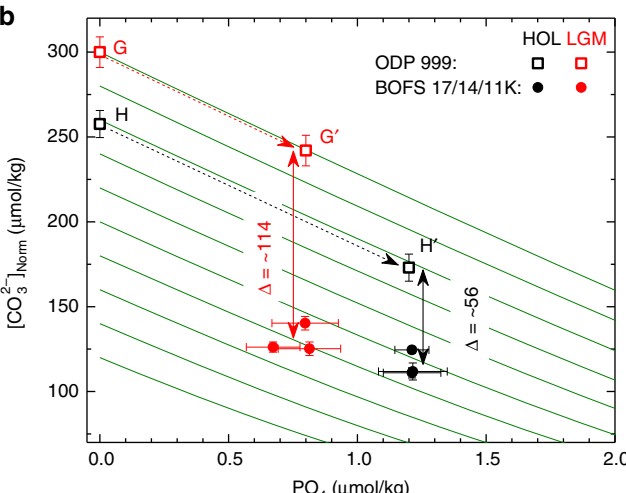

**Fig. 5** $[CO_3^{2-}]_{Norm}$ vs. $PO_4$. **a** Preindustrial Atlantic surface (<100 m, north of 10°N) and deep (>1000 m, 65°N–65°S) water data[2]. **b** Holocene and LGM data. Error bars, $2\sigma$

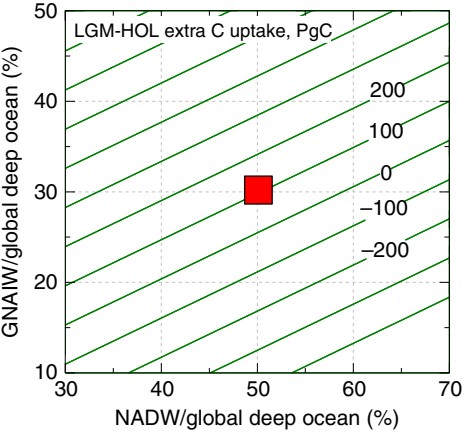

**Fig. 6** North Atlantic $CO_2$ budget. The LGM–Holocene extra carbon uptake is based on Holocene-to-LGM $DIC_{as}$ increase of 91 µmol/kg. The large red square represents our best estimate of ~100 PgC, assuming that NADW and GNAIW occupied ~50% and ~30% of the global deep ocean (>1 km), respectively[35,36,38,39]. See Methods for calculation details

~2 km depth (BOFS 11 K core depth) in the glacial North Atlantic.

**Quantification of North Atlantic $CO_2$ uptake.** With reconstructed ODP 999−BOFS $[CO_3^{2-}]_{as}$ gradients, we further quantify North Atlantic air–sea $CO_2$ absorption changes between the Holocene and LGM. $[CO_3^{2-}]_{as}/DIC_{as}$ sensitivities can be precisely estimated (Fig. 4e), making $[CO_3^{2-}]_{as}$ gradients a useful proxy to calculate $DIC_{as}$ changes. The 58 ± 12 µmol/kg Holocene-to-LGM $[CO_3^{2-}]_{as}$ increase (Fig. 5b) indicates a $DIC_{as}$ increase of 91 ± 20 µmol/kg due to enhanced North Atlantic air–sea $CO_2$ absorption (Methods). Compared to the preindustrial Gulf Stream-NADW $DIC_{as}$ gradient of ~90 µmol/kg (Fig. 2, Supplementary Fig. 3), this suggests a doubling of $CO_2$ uptake efficiency in the LGM North Atlantic.

Beside $DIC_{as}$ gradient changes, which indicate air–sea $CO_2$ uptake efficiency, knowledge of northern-sourced-water volumes in the global deep ocean is required to determine total North Atlantic carbon sequestration. Figure 6 shows the total extra carbon absorbed by the LGM North Atlantic for a range of northern-sourced-water volumes (Methods). Sedimentary Pa/Th, radiocarbon, neodymium isotopes, and paired benthic Cd/Ca–$\delta^{13}C$ suggest[32,34,35,37] vigorous glacial northern-sourced intermediate water production and subsequent transport to the remaining world ocean. Based on previous estimates[35,36,38,39], we tentatively assume that NADW- and GNAIW-derived waters occupy ~50% and ~30%, respectively, of the global deep ocean volume ($1 \times 10^{18}$ m³ for >1 km). In this case, our ~91 µmol/kg Holocene-to-LGM $DIC_{as}$ increase yields ~100 Petagrams of carbon (PgC; 1 Pg = $1 \times 10^{15}$ g) greater $CO_2$ sequestration by the LGM North Atlantic (Fig. 6; Methods). To maintain similar total carbon uptake between the Holocene and LGM, GNAIW would need to be less than ~50% of NADW in volume, which we consider unlikely given evidence for intensive GNAIW export to the global ocean[23,32,34,35,37]. We acknowledge uncertainties associated with our calculations, and encourage future work to better constrain volumes and carbonate chemistry changes of various water masses in the past.

**Discussion**

Previous work[40–42] has tried to constrain air–sea $CO_2$ exchange by reconstructing surface conditions. This requires reconstructions of the air–sea $pCO_2$ difference (influenced by T, S, and nutrient

would raise the LGM $[CO_3^{2-}]_{as}$ gradient (Supplementary Figs. 16 and 17).

Regarding ocean circulation changes, most AAIW upwells in the tropics and less than ~25% of today's NADW is fed directly by AAIW without surfacing at low latitudes[17]. Northward AAIW transport is thought to have been reduced substantially in the glacial Atlantic[23,31–33] in the face of vigorous GNAIW production[34]. Assuming a constant total carbon uptake by the North Atlantic, a complete shutdown of AAIW contribution would only raise the ODP 999−BOFS $[CO_3^{2-}]_{as}$ gradient by ~30%, which is much smaller than the ~100% increase from the Holocene (~56 µmol/kg) to LGM (~114 µmol/kg) (Fig. 5b). Any increased mixing of glacial AABW at BOFS sites would reduce the ODP 999−BOFS $[CO_3^{2-}]_{as}$ gradient during the LGM. Given the proximity of our deep-water sites to the core of GNAIW and Nordic Sea overflow waters[23,24,31,35,36], the larger LGM $[CO_3^{2-}]_{as}$ gradient between ODP 999 and BOFS cores likely reflects a greater $DIC_{as}$ increase from Gulf Stream to GNAIW. North Atlantic $CO_2$ invasion was responsible for the preindustrial Gulf Stream-NADW $[CO_3^{2-}]_{as}$ gradient (Fig. 2). Therefore, we ascribe the increased ODP 999−BOFS $[CO_3^{2-}]_{as}$ gradient during the LGM to more efficient atmospheric $CO_2$ uptake via air–sea exchange and subsequent transport to at least

utilization), the gas transfer velocity (a power function of wind speed), solubility of $CO_2$ in seawater (mainly affected by T), and the area and contact time of surface waters available for air–sea exchange[1]. Sea ice cover[43] possibly expanded, reducing glacial North Atlantic $CO_2$ absorption. A larger LGM meridional surface temperature gradient[43,44] would enhance the North Atlantic solubility pump[13]. Existing planktonic $\delta^{15}N$ and Cd/Ca data[40,45] show conflicting results regarding the glacial North Atlantic nutrient conditions, perhaps due to complications associated with surface-water proxies and spatial/seasonal nutrient variations in the North Atlantic. A decreased preformed nutrient in the glacial North Atlantic might be inferred from a lower GNAIW $PO_4$ (Fig. 3), but faster ventilation and/or reduced glacial AAIW could also cause a nutrient decline in GNAIW[23,34,46]. Little is known about past wind intensity and air–sea contact time changes. Consequently, potential North Atlantic glacial $CO_2$ invasion remains poorly understood. Bypassing the necessity to reconstruct surface-water conditions for which some proxies are still lacking (e.g., wind), our new approach, to our knowledge, offers the first proxy-based quantitative estimate of air–sea $CO_2$ uptake efficiency in the glacial North Atlantic.

In contrast to previous calculations[47–49] which concern combined biological (i.e., within-ocean DIC redistribution) and air–sea exchange carbon changes (Fig. 1c), our total North Atlantic carbon uptake estimate only represents the net air–sea $CO_2$ change that is more directly relevant to atmospheric and terrestrial carbon inventory variations. Our estimated ~100 PgC sequestration constitutes ~15% of the Holocene-LGM ~600 PgC change associated with the atmosphere (~200 PgC) and terrestrial biosphere (~400 PgC)[5,6]. Given this global carbon budget context, our work reinforces the role of other polar regions (e.g., Southern Ocean) in controlling the glacial–interglacial carbon cycle. However, if there were no efficiency enhancement for the LGM North Atlantic, a 40% shrinkage of NADW volume would decrease air–sea component $CO_2$ sequestration by ~240 PgC in the deep ocean (Methods). Therefore, by overcoming this opposing "volume effect", the improved glacial North Atlantic efficiency increased $DIC_{as}$ values of northern-sourced deep waters (termed the "endmember effect") and thereby contributed substantially to air–sea $CO_2$ sequestration in the LGM deep ocean.

Atmospheric $pCO_2$ is controlled by both $CO_2$ gains (e.g., via Southern Ocean outgassing) and losses (e.g., via North Atlantic absorption)[2,3,11]. Growing evidence indicates that processes outside the Southern Ocean may have affected past atmospheric $CO_2$ variations[50–52]. Our proxy-based results indicate that the North Atlantic $CO_2$ pump efficiency during the LGM was almost doubled relative to the Holocene. This increased efficiency and associated "endmember effect" effectively outcompeted the opposing "volume effect" due to any shrinkage of northern-sourced deep waters in the world ocean. In addition to the well-recognized role of reduced outgassing in the Southern Ocean[6,8,9,47,53,54], we therefore suggest that variations in the uptake and sequestration of atmospheric $CO_2$ via the North Atlantic Ocean were important contributors to glacial/interglacial carbon cycling.

## Methods

**$CO_2$ system calculations**. For both the preindustrial ocean and down-core $CO_2$ system calculations, seawater carbonate system variables were calculated using the $CO_2$sys.xls program[28] with dissociation constants $K_1$ and $K_2$ according to Mehrbach et al.[55] and $K_{SO_4}$ according to Dickson[56]. Seawater total boron concentration was calculated from the boron–salinity relationship of Lee et al.[57]. For the GLODAP dataset, the anthropogenic $CO_2$ contribution was subtracted from the measured DIC to obtain preindustrial DIC values[2].

**Preindustrial Atlantic $DIC_{as}$ and $[CO_3^{2-}]_{as}$**. The GLODAP dataset[2] is used to calculate preindustrial ocean $CO_2$ system variables. Following the established

method of Broecker and Peng[3], we account for DIC anomalies created by (1) freshwater addition or removal based on S, (2) soft-tissue carbon creation and respiration based on $PO_4$, and (3) $CaCO_3$ formation and dissolution based on ALK and nitrate ($NO_3$). See Fig. 1 for the simplified concept. We adopt the term $DIC_{as}$ to represent net air–sea exchange component DIC signatures from:

$$DIC_{as} = DIC_s - (PO_{4s} - PO_4^{mo}) \times C/PO_4$$
$$- \tfrac{1}{2} \times (ALK_s - ALK^{mo} + NO_{3s} - NO_3^{mo}) - DIC_{constant}$$
(1)

where the subscript "s" represents values normalized to S of 35 (e.g., $DIC_s = DIC \times 35/S$); the superscript "mo" denotes mean ocean values at $S = 35$ ($PO_4^{mo} = 2.2$ μmol/kg, $ALK^{mo} = 2383$ μmol/kg, $DIC^{mo} = 2267$ μmol/kg, and $NO_3^{mo} = 31$ μmol/kg)[29]; $C/PO_4$ represents the soft-tissue stoichiometric Redfield ratio; and the arbitrary $DIC_{constant}$ ($= 2285$ μmol/kg) is designed to bring zero $DIC_{as}$ close to the NADW–AABW boundary (Fig. 2). The term $(PO_{4s} - PO_4^{mo}) \times C/PO_4$ corrects for DIC changes due to photosynthesis and soft-tissue degradation, and the term $\tfrac{1}{2} \times (ALK_s - ALK^{mo} + NO_{3s} - NO_3^{mo})$ accounts for DIC changes caused by $CaCO_3$ formation and dissolution. To be consistent with previous work[3,30], we used $C/PO_4 = 127$ to calculate $DIC_{as}$ and $[CO_3^{2-}]_{as}$ in Fig. 2. Using other $C/PO_4$ values[10] does not significantly affect spatial $DIC_{as}$ and $[CO_3^{2-}]_{as}$ patterns (Supplementary Figs. 11 and 12). Neither are their patterns affected by using other $PO_4$–ALK–$NO_3$ values to replace global mean values in Eq. (1) (Supplementary Figs. 13 and 14). Ideally, $DIC_{constant}$ would be the mean DIC value of an abiotic ocean (Fig. 1), but this value cannot be simply determined from modern observations. Because our interest lies in spatial $DIC_{as}$ contrasts instead of absolute values, the choice of $DIC_{constant}$ has no effect on our interpretation.

To obtain $[CO_3^{2-}]_{as}$, we first calculate $[CO_3^{2-}]_{PO4-T-S-P}$ using ($DIC_{as} + DIC_{constant}$), $ALK^{mo}$, and $PO_4^{mo}$ at $T = 3$ °C, $S = 35$, and $P = 2500$ dbar. $[CO_3^{2-}]_{as}$ is then calculated by $[CO_3^{2-}]_{as} = [CO_3^{2-}]_{PO4-T-S-P} - [CO_3^{2-}]_{constant}$, where $[CO_3^{2-}]_{constant}$ ($= 78$ μmol/kg, calculated using $DIC_{constant}$ and $ALK^{mo}$) is designed to bring zero $[CO_3^{2-}]_{as}$ close to the NADW–AABW boundary. In essence, the $[CO_3^{2-}]_{as}$ distribution reflects the variation of $[CO_3^{2-}]$ when normalized to the same $PO_4-T-S-P$ conditions.

**$CO_2$ system sensitivities and calculation of $[CO_3^{2-}]_{Norm}$**. Because the seawater $CO_2$ system is nonlinear, there is currently no simple way to derive these sensitivities based on $CO_2$ system equations[16]. We use GLODAP preindustrial data[2] to calculate numerically $[CO_3^{2-}]$ sensitivities to various physiochemical parameters. Use of LGM outputs from the LOVECLIM model[58] yields comparable sensitivities. We first use hydrographic data, including T, S, P, DIC, ALK, $PO_4$, and $SiO_3$ to calculate $[CO_3^{2-}]$. We then change S to 35‰ and other chemical concentrations proportionally. For example, ALK and DIC will change as follows:

$$ALK_{s=35} = ALK \times 35/S, \text{ and}$$
(2)

$$DIC_{s=35} = DIC \times 35/S.$$
(3)

We use $S = 35‰$, $ALK_{S=35}$, $DIC_{S=35}$, $[PO_4]_{S=35}$, and $[SiO_3]_{S=35}$ along with hydrographic T and P to calculate $[CO_3^{2-}]_{S=35}$. The $[CO_3^{2-}]$ to S sensitivity ($Sen_S$) is calculated by:

$$Sen_S = \left([CO_3^{2-}] - [CO_3^{2-}]_{S=35}\right)/(S - 35).$$
(4)

To estimate temperature effects, we calculate $[CO_3^{2-}]_{S=35, T=3 °C}$ using $S = 35‰$, $ALK_{S=35}$, $DIC_{S=35}$, $[PO_4]_{S=35}$, $[SiO_3]_{S=35}$, $T = 3$ °C, and hydrographic P. The sensitivity of $[CO_3^{2-}]_{S=35}$ to temperature ($Sen_T$) is defined by:

$$Sen_T = \left([CO_3^{2-}]_{S=35, T=3°C} - [CO_3^{2-}]_{S=35}\right)/(3 - T).$$
(5)

Regarding pressure effects, we calculate $[CO_3^{2-}]_{S=35, T=3 °C, P=2500 dbar}$ using $S = 35‰$, $ALK_{S=35}$, $DIC_{S=35}$, $[PO_4]_{S=35}$, $[SiO_3]_{S=35}$, $T = 3$ °C, and $P = 2500$ dbar. The sensitivity of $[CO_3^{2-}]_{S=35, T=3°C}$ to P ($Sen_P$) is defined by:

$$Sen_P = \left([CO_3^{2-}]_{S=35, T=3°C, P=2500\,dbar}\right.$$
$$\left. - [CO_3^{2-}]_{S=35, T=3°C}\right)/(2500 - P) \times 100.$$
(6)

To estimate the influence on $[CO_3^{2-}]$ from within-ocean ALK–DIC redistributions by biological processes, we assume a 0.1 μmol/kg increase in $PO_4$ (i.e., $\Delta PO_4 = 0.1$ μmol/kg) due to biological respiration (photosynthesis has an opposite effect). The resultant ALK ($ALK_{S=35+respiration}$) and DIC ($DIC_{S=35+respiration}$) can then be calculated from:

$$ALK_{S=35+respiration} = ALK_{S=35} + \Delta PO_4 \times C/PO_4$$
$$\div R \times 2 - \Delta PO_4 \times N/PO_4.$$
(7)

$$DIC_{S=35+respiration} = DIC_{S=35} + \Delta PO_4 \times C/PO_4 + \Delta PO_4 \times C/PO_4 \div R.$$
(8)

Resultant $[CO_3^{2-}]$ ($[CO_3^{2-}]_{Norm+respiration}$) values are calculated using $DIC_{S=35+respiration}$, $ALK_{S=35+respiration}$, and ($[PO_4]_{S=35} + \Delta PO_4$) at constant physical conditions of $T = 3$ °C, $S = 35$, and $P = 2500$ dbar. The sensitivity of

$[CO_3^{2-}]_{Norm}$ to $PO_4$ is defined by:

$$[CO_3^{2-}]_{Norm}/PO_4 \text{ sensitivity} =$$
$$\left([CO_3^{2-}]_{Norm+respiration} - [CO_3^{2-}]_{Norm}\right)/\Delta PO_4. \quad (9)$$

We consider four Redfield stoichiometric scenarios: $C/PO_4 = 127$, $R = 4$ (the reference composition; Fig. 4d); $C/PO_4 = 140$, $R = 4$; $C/PO_4 = 127$, $R = 8$; and $C/PO_4 = 140$, $R = 8$ (Supplementary Fig. 15). In all cases, strong exponential correlations exist between $[CO_3^{2-}]_{Norm}/PO_4$ sensitivities and $[CO_3^{2-}]_{Norm}$. The correlations may reflect the buffering effect of the seawater $CO_2$ system: for seawater with high DIC (low $[CO_3^{2-}]$ and high buffering capability), $[CO_3^{2-}]$ would be relatively less sensitive to biological DIC and ALK disturbances. All of the above sensitivity calculations assume no air–sea $CO_2$ change.

To calculate air–sea exchange sensitivities, we assume a 10 μmol/kg increase in $DIC_{S=35}$ due to atmospheric $CO_2$ invasion (i.e., $\Delta DIC_{as} = 10$ μmol/kg). We calculate $[CO_3^{2-}]_{Norm+as}$ using $S = 35‰$, $ALK_{S=35}$, $DIC_{S=35+as}$ ($= DIC_{S=35} + \Delta DIC_{as}$), $[PO_4]_{S=35}$, $[SiO_3]_{S=35}$, $T = 3\,°C$, and $P = 2500$ dbar. The sensitivity of $[CO_3^{2-}]_{as}$ to $DIC_{as}$ is defined by:

$$[CO_3^{2-}]_{as}/DIC_{as} \text{ sensitivity} =$$
$$\left([CO_3^{2-}]_{Norm+as} - [CO_3^{2-}]_{Norm}\right)/\Delta DIC_{as}. \quad (10)$$

Using sensitivities shown in Fig. 4, $[CO_3^{2-}]_{Norm}$ can be calculated by:

$$[CO_3^{2-}]_{Norm} = [CO_3^{2-}] + (35 - S) \times Sen_{\_S} + (3 - T)$$
$$\times Sen_{\_T} + (2500 - P)/100 \times Sen_{\_P}. \quad (11)$$

Excel spreadsheets are provided in Supplementary Data 7–8 to calculate $[CO_3^{2-}]_{Norm}$ and the biological curves shown in Fig. 5.

**LGM–Holocene North Atlantic carbon budget.** The total extra carbon increase ($\Delta\Sigma C_{LGM-Holocene}$) in Fig. 6 is calculated by $\Delta\Sigma C_{LGM-Holocene} = V \times density \times \%GNAIW \times ([CO_3^{2-}]_{as\_ODP999-BOFS}^{LGM}/0.61) \times 12 - V \times density \times \%NADW \times ([CO_3^{2-}]_{as\_ODP999-BOFS}^{Holocene}/0.59) \times 12$, where $V$ is the global deep ocean volume (>1 km water depth) at $100.8 \times 10^{16}$ m$^3$, density = 1027.8 kg/m$^3$ (ref. [29]), %GNAIW and %NADW, respectively, represent their volume fractions in the deep ocean, $[CO_3^{2-}]_{as\_ODP999-BOFS}^{Holocene} = 56$ μmol/kg, $[CO_3^{2-}]_{as\_ODP999-BOFS}^{LGM} = 114$ μmol/kg (Fig. 5), terms 0.61 and 0.58, respectively, represent the absolute LGM and Holocene $[CO_3^{2-}]_{as}/DIC_{as}$ sensitivities (Fig. 4e) used to transfer $[CO_3^{2-}]_{as\_ODP999-BOFS}$ into ODP999–BOFS $DIC_{as}$ contrasts (LGM: 186 μmol/kg; Holocene: 95 μmol/kg), and the number 12 converts C from moles into weight. Based on previous estimates, %NADW is thought to be ~50% (refs. [38,39]), while %GNAIW remained roughly similar to %NADW or shrank (refs. [35,36]). These estimates are debated and have large uncertainties, and we thus calculate $\Delta\Sigma C_{LGM-Holocene}$ for a range of %NADW and %GNAIW values (Fig. 6). Any influence from AAIW is ignored because of its similar $[CO_3^{2-}]_{as}$ signals to Gulf Stream during the Holocene (Supplementary Fig. 3) and much reduced northward advection during the LGM[23,31–33]. We tentatively treat $\Delta\Sigma C_{LGM-Holocene}$ of ~100 PgC using %NADW = 50% and %GNAIW = 30% as our best estimate. Assuming no Holocene–LGM $DIC_{as}$ gradient change (i.e., the same $CO_2$ uptake efficiency) and everything else being equal, $\Delta\Sigma C_{LGM-Holocene}$ would be −240 PgC at %NADW = 50% and %GNAIW = 30%.

**Cores, age models, samples, and analytical methods.** We used ODP Site 999 for Gulf Stream surface-water reconstructions (Fig. 2). The age model is from Schmidt et al.[59]. Planktonic foraminiferal *Globigerinoides ruber* (*sensu stricto*, white variety) $\delta^{18}O$, Mg/Ca, and $\delta^{11}B$ data are from refs. [21,22,59]. Briefly, about 25 and 55 shells from the 250–350 μm size fraction were used for $\delta^{18}O$ and Mg/Ca analyses, respectively. Samples for $\delta^{18}O$ analyses were sonicated in methanol for 5–10 s, roasted under vacuum at 375 ºC for 30 min, and analyzed on a Fisons Optima IRMS with a precision of <0.06‰. Shells for Mg/Ca were cleaned following the reductive cleaning procedure[60] and measured on an inductively-coupled plasma mass spectrometer (ICP-MS) with a precision of ~1.7%. For $\delta^{11}B$ analyses, about 100–120 *G. ruber* (w) shells from the 300–355 μm size fraction were cleaned following the "Mg-cleaning" procedure[61], to minimize material loss during cleaning[62]. *G. ruber* (w) $\delta^{11}B$ was measured on a Neptune multicollector (MC)–ICP-MS with an analytical error in $\delta^{11}B$ of about ±0.25‰ (ref. [21]).

Three cores (BOFS 17, BOFS 11, and BOFS 14 K) from the polar North Atlantic Ocean are used for deep-water reconstructions (Fig. 3). Their age models are based on published chronologies[24,63–65]. For each sample (~2 cm thickness), ~10–20 cm$^3$ of sediment was disaggregated in de-ionized water and was wet sieved through 63 μm sieves. To facilitate analyses, we picked the most abundant species for measurements. For each Cd/Ca analysis, ~10–20 monospecific shells of the benthic foraminifera *C. mundulus* (BOFS 17 K) and *C. wuellerstorfi* (BOFS 14, 11 K) were obtained from 250 to 500 μm size fraction. The shells were double checked under a microscope before crushing to ensure that consistent morphologies were used throughout the core. On average, following this careful screening the starting material for each sample was ~8–12 shells, which is equivalent to ~300–600 μg of carbonate. For benthic B/Ca analyses, foraminiferal shells were cleaned with either the "Mg-cleaning" method[61] or the "Cd-cleaning" protocol[61], to investigate

cleaning effects on trace element/Ca in foraminiferal shells[62,66]. No discernable B/Ca difference is observed between the two cleaning methods[25,62]. Benthic B/Ca ratios were measured on an ICP-MS using procedures outlined in ref. [67], with an analytical error better than ~5%.

For each benthic Cd/Ca analysis, ~10–20 shells of the benthic foraminiferal taxa *C. mundulus* (BOFS 17 K), *C. wuellerstorfi* (BOFS 14, 11 K), and *Uvigerina* spp. (BOFS 17 K) were picked from the 250–500 μm size fraction. Previous studies[26,27,68] showed similar Cd/Ca ratios between infaunal *Uvigerina* spp. and epifaunal *Cibicidoides*, and we thus combined Cd/Ca data from these taxa to obtain continuous downcore $PO_4$ records. We used the "Cd-cleaning" method[60,69] to clean benthic shells for Cd/Ca measurements. Cd/Ca ratios were measured on an ICP-MS with an analytical error better than ~5% (ref. [67]).

For $\delta^{11}B$ measurements, about 20 benthic shells from the 250–500 μm size fraction were picked for each sample. Shells used for $\delta^{11}B$ analyses were cleaned using the "Mg-cleaning" method[61], to minimize loss of shell material[61]. After cleaning, shells were dissolved and pure boron was extracted using column chemistry as described by Foster[21]. Benthic $\delta^{11}B$ was measured on a Neptune multi-collector (MC)–ICP-MS following ref. [21]. The analytical error in $\delta^{11}B$ is about ± 0.25‰. Due to the relatively large sample size requirement, shell availability, and lengthy chemical treatments for $\delta^{11}B$, we present low-resolution $\delta^{11}B$ for *C. mundulus* from BOFS 17 K and for *C. wuellerstorfi* from BOFS 11 K. Note that consistent $[CO_3^{2-}]$ results from B/Ca and $\delta^{11}B$ strengthen the reliability of our reconstructions (Fig. 3).

Published benthic Cd/Ca and B/Ca results are included in Fig. 3. Altogether, we generated 180 new measurements of benthic $\delta^{11}B$, B/Ca, and Cd/Ca. All data are listed in Supplementary Data 1–9.

**ODP 999 reconstructions.** ODP Site 999 was used to constrain past physical conditions and carbonate chemistry of the Gulf Stream (Supplementary Fig. 4). Following previous approaches[21,22], surface water temperature ($T_{surface}$) and salinity ($S_{surface}$) were estimated based on *G. ruber* Mg/Ca (ref. [59]) and sea level changes[21,22,59], respectively. We first convert *G. ruber* $\delta^{11}B$ to borate $\delta^{11}B$ ($\delta^{11}B_{borate}$), following the conversion method of ref. [22]. Surface water pH ($pH_{surface}$) was calculated from seawater $\delta^{11}B_{borate}$ along with $T_{surface}$ and $S_{surface}$. To constrain the $CO_2$ system, two $CO_2$ system variables are necessary[16]. In addition to $\delta^{11}B$-derived pH, literature studies[21,22,41] generally estimate past surface-water ALK ($ALK_{surface}$) changes. Following refs. [21,22], we estimate $ALK_{surface}$ from $S_{surface}$ using the modern $S_{surface}-ALK_{surface}$ relationship ($ALK_{surface} = 59.19 \times S_{surface} + 229.08$, $R^2 = 0.99$)[21]. Together with $T_{surface}$ and $S_{surface}$, $pH_{surface}$, and $ALK_{surface}$ were used to calculate other $CO_2$ system variables including surface-water $[CO_3^{2-}]$ ($[CO_3^{2-}]_{surface}$) and DIC ($DIC_{surface}$) using the $CO_2$sys program[28]. Surface-water $PO_4$ concentration at ODP 999 is assumed to be zero over the last 27 ka.

Following refs. [21,22,59], errors are estimated to be 1 °C, 1‰, 100 μmol/kg, and ~0.43‰ for $T_{surface}$, $S_{surface}$, $ALK_{surface}$, and $\delta^{11}B_{borate}$, respectively. Integrated average uncertainties in $[CO_3^{2-}]_{surface}$ and $DIC_{surface}$ for a single reconstruction are, respectively, ~20 (Holocene: ~18, LGM: ~24) and ~90 μmol/kg, based on quadratic addition of all individual errors sourced from $T_{surface}$ ($[CO_3^{2-}]_{surface}$: 2 μmol/kg, $DIC_{surface}$: 3 μmol/kg), $S_{surface}$ ($[CO_3^{2-}]_{surface}$: 2 μmol/kg, $DIC_{surface}$: 5 μmol/kg), $ALK_{surface}$ ($[CO_3^{2-}]_{surface}$: 14 μmol/kg, $DIC_{surface}$: 86 μmol/kg), and $\delta^{11}B_{borate}$ ($[CO_3^{2-}]_{surface}$: 16 μmol/kg, $DIC_{surface}$: 24 μmol/kg; note that $\delta^{11}B_{borate}$ leads to an error in $[CO_3^{2-}]$ via pH). Uncertainties for calculated $CO_2$ system variables at ODP 999 are tabulated in Supplementary Data 1. Use of other methods to estimate ALK would have little impact on our conclusions (Supplementary Figs. 16 and 17).

**From pH to $[CO_3^{2-}]$.** For palaeo-studies, surface-water pH is generally obtained from planktonic foraminiferal $\delta^{11}B$. To calculate $[CO_3^{2-}]$, a second $CO_2$ system variable is needed[16]. Following the previous approach[21,22], past $ALK_{surface}$ at ODP 999 has been estimated from S using the $S_{surface}-ALK_{surface}$ relationship. Due to limited knowledge about the past $S_{surface}-ALK_{surface}$ relationship, a generous uncertainty has been assigned to $ALK_{surface}$ at ±100 μmol/kg (ref. [21,22]), which is about half of the entire ALK range in the present global ocean[2]. Using $ALK_{surface}$ and $pH_{surface}$ along with $T_{surface}$ and $S_{surface}$, $[CO_3^{2-}]_{surface}$ and $DIC_{surface}$ can be calculated using the $CO_2$sys program[28]. Because of the large uncertainty in $ALK_{surface}$, large errors in $DIC_{surface}$ might be expected (Supplementary Fig. 4). However, given the constraint from $pH_{surface}$, seawater $ALK_{surface}$ and $DIC_{surface}$ variations are not random but must vary systematically within ALK–DIC space (Supplementary Fig. 5). Because of the close relationship between pH and $[CO_3^{2-}]$ (i.e., roughly parallel patterns of pH and $[CO_3^{2-}]$ within ALK–DIC space; Supplementary Fig. 5), this systematic ALK–DIC variation allows us to confine $[CO_3^{2-}]$ with acceptable uncertainty. For a given pH at ODP 999, an error of 100 μmol/kg in ALK only leads to an error of about ±14 μmol/kg in $[CO_3^{2-}]$ (Supplementary Fig. 5).

For clarity, Supplementary Fig. 5a, b only consider the effect of ALK errors on $[CO_3^{2-}]$ estimates assuming constant pH and T–S–P conditions. To fully propagate errors from various sources including $T_{surface}$, $S_{surface}$, $ALK_{surface}$, and $pH_{surface}$, we use a Monte Carlo approach ($n = 10,000$) to calculate the integrated error in $[CO_3^{2-}]$ (ref. [70]). As can be seen from Supplementary Fig. 5c–f, the final errors (~20–25 μmol/kg) in an individual $[CO_3^{2-}]$ reconstruction based on the Monte-Carlo are similar to those (~18–24 μmol/kg) based on quadratic addition of individual errors, justifying our major error estimation approach (i.e., quadratic addition).

**Subtropical western North Atlantic surface [CO₃²⁻].** Because most of North Atlantic subtropical gyre waters circulate through the Caribbean Sea before being transported to the subpolar North Atlantic via the Gulf Stream, ODP 999 from Caribbean Sea is used to constrain past Gulf Stream carbonate chemistry[20]. To further test the feasibility of using ODP 999 to represent the first-order Gulf Stream [CO₃²⁻] changes during the Holocene and LGM, we have estimated surface-water [CO₃²⁻] for four sites from the wider subtropical western Atlantic region (latitude: 12–33°N, longitude: 61–91°W). Among these sites, KNR140–51GGC (33°N, 76°W) is located within the Gulf Stream today[71]. Because subtropical surface waters cycle multiple times through the upper ocean gyre circulations, it is possible that surface waters have been close to equilibrium with past atmospheric $pCO_2$ (refs. [21,22]). Therefore, we assume surface-water $pCO_2$ of 270 and 194 ppm for the Holocene and LGM, respectively[72]. We assign a ±15 ppm error to surface-water $pCO_2$ to account for any potential air–sea $CO_2$ disequilibrium. For these sites, we use surface temperature and salinity reconstructions from previous publications[71,73–75]. ALK is calculated based on the same approach for ODP 999. The reconstructed in situ [CO₃²⁻] values show some differences between cores, due to local T–S conditions. Since we are interested in air–sea $CO_2$ exchange signals, we convert reconstructed in situ [CO₃²⁻] into [CO₃²⁻]_Norm using Eq. (11). As can be seen from Supplementary Fig. 6 and Supplementary Data 2, these cores show similar [CO₃²⁻]_Norm values for the Holocene (~260 μmol/kg) and LGM (~300 μmol/kg) as ODP 999. Therefore, we argue that ODP 999 sufficiently records first-order Gulf Stream air–sea exchange carbonate chemistry for the Holocene and LGM. Because we aim to obtain a proxy-based estimates, we use ODP 999 data for calculations in the main text.

**Benthic B/Ca and δ¹¹B to deep-water [CO₃²⁻].** Most deep-water [CO₃²⁻] values are reconstructed using benthic B/Ca (refs. [25,47]) from $[CO_3^{2-}]_{downcore} = [CO_3^{2-}]_{PI} + \Delta B/Ca_{downcore-coretop}/k$, where [CO₃²⁻]_PI is the preindustrial (PI) deep-water [CO₃²⁻] value estimated from the GLODAP dataset[2], $\Delta B/Ca_{downcore-coretop}$ represents the deviation of B/Ca of down-core samples from the core-top value, and $k$ is the B/Ca–[CO₃²⁻] sensitivity of *C. wuellerstorfi* (1.14 μmol/mol per μmol/kg) or *C. mundulus* (0.69 μmol/mol per μmol/kg)[25]. We use a reconstruction uncertainty of ±10 μmol/kg in [CO₃²⁻] based on global core-top calibration samples[25,76].

For cores BOFS 17 K and BOFS 11 K, new monospecific epifaunal benthic δ¹¹B values were converted into deep-water [CO₃²⁻] following the approach detailed in ref. [77]. Briefly, benthic δ¹¹B is assumed to directly reflect deep-water borate δ¹¹B, as suggested by previous core-top calibration work[78]. Deep-water pH is calculated using benthic δ¹¹B along with $T_{deep}$ and $S_{deep}$, similar to the approach to calculate surface-water pH at ODP 999 (refs. [21,22]). We assume constant ALK at the studied sites (2313 μmol/kg at BOFS 17 K and 2310 μmol/kg at BOFS 11 K) in the past. Following ref. [77], a generous error of 100 μmol/kg is assigned to ALK estimates. We then calculate deep-water [CO₃²⁻] from pH and ALK using the CO2sys program[28]. The integrated average uncertainty in deep-water [CO₃²⁻] is ~ ±10 μmol/kg, based on quadratic addition of individual errors of ~±2 μmol/kg sourced from $T_{deep}$ (±1 °C), ~±2 μmol/kg from $S_{deep}$ (±1‰), ~±5 μmol/kg from ALK (±100 μmol/kg), and ~±8 μmol/kg from δ¹¹B_borate (~±0.25‰). As demonstrated by Supplementary Fig. 5, the large ALK error only contributes a small uncertainty to the final [CO₃²⁻] estimate. As shown in Fig. 3, benthic B/Ca and δ¹¹B yield consistent deep-water [CO₃²⁻] reconstructions for the Holocene and LGM.

**Benthic Cd/Ca to deep-water PO₄.** We follow the established approach[26,46,79] to convert benthic (*C. wuellerstorfi*, *C. mundulus*, and *Uvigerina* spp.) foraminiferal Cd/Ca into deep-water Cd concentrations. Partition coefficients ($D_{Cd}$) are used to calculate deep water Cd from: $Cd$ (nmol/kg) = $[(Cd/Ca)_{foram}/D_{Cd}] \times 10$. Bertram et al.[65] used empirical $D_{Cd}$ values of 2.3, 2.2, and 2.7 for BOFS 17, 14, and 11 K, respectively. However, these $D_{Cd}$ values would result in Holocene Cd of 0.3–0.4 nmol/kg, higher than the observed value of ~0.25 nmol/kg from modern hydrographic measurements (Supplementary Fig. 7)[80]. This offset may suggest higher $D_{Cd}$ values for the North Atlantic Ocean, which has been acknowledged recently[81]. We thus adjust $D_{Cd}$ (~25% increase) so that the calculated Holocene deep-water Cd concentrations match modern measurements. This adjustment is supported by consistent Cd reconstructions from this study and previous reconstructions based on Cd/Ca measurements for *Hoeglundina elegans*. Compared to *Cibicidoides*, $D_{Cd}$ into *H. elegans* is far less variable[79]. As can be seen from Supplementary Fig. 8, for cores with similar benthic δ¹³C from similar water depths (i.e., bathed in similar water masses), our Cd reconstructions match favorably with those based on *H. elegans* measurements[82]. Deep water Cd is converted into PO₄ using the relationship based on the latest North Atlantic Ocean measurements (Supplementary Fig. 7)[80]. Using older published Cd–PO₄ relationships[26,83] only marginally affects our PO₄ estimates.

Uncertainties associated with Cd and PO₄ reconstructions are estimated as follows. Error for Cd is estimated using $2\sigma_{Cd} = \sqrt{(2\sigma_{D_{Cd}})^2 + (2\sigma_{Cd/Ca})^2}$, where $2\sigma_{D_{Cd}}$ and $2\sigma_{Cd/Ca}$ (=5%) are errors for $D_{Cd}$ and Cd/Ca, respectively. Due to poorly defined uncertainty for $D_{Cd}$ from the literature, we assume an error of 50%, and then compare our final errors with literature estimates to assess the appropriateness of our calculations. Seawater PO₄ is calculated from Cd using: $PO_4 = \frac{Cd - b \pm (2\sigma_b)}{a \pm (2\sigma_a)}$,

where $2\sigma_a$ and $2\sigma_b$, respectively, represent 95% confidence errors associated with $a$ and $b$ (Supplementary Fig. 7b). The PO₄ uncertainty was calculated from:

$$2\sigma_{PO_4} = \sqrt{\left(\partial_{PO_4}/\partial_a \cdot 2\sigma_a\right)^2 + \left(\partial_{PO_4}/\partial_b \cdot 2\sigma_b\right)^2 + \left(\partial_{PO_4}/\partial_{Cd} \cdot 2\sigma_{Cd}\right)^2}, \text{ where}$$

$\partial_{PO_4}/\partial_a = \frac{-(Cd-b)}{a^2}$, $\partial_{PO_4}/\partial_b = \frac{-1}{a}$, and $\partial_{PO_4}/\partial_{Cd} = \frac{1}{a}$. Our final errors on individual Cd and PO₄ are ~0.12 nmol/kg (~55%) and ~0.5 μmol/kg (~50%), respectively. When compared with previously published uncertainties (~0.08 nmol/kg for Cd and ~0.17 μmol/kg for PO₄)[46,68], our error estimates are possibly too generous. Here we use ~50% error to be conservative. We encourage future work to improve uncertainty estimates for the benthic Cd/Ca proxy.

The oceanic residence time of PO₄ is ~100,000 years[84]. The LGM deep ocean was possibly more reducing[85], which might have facilitated sediment organic matter preservation, and, thus, PO₄ removal from the ocean. However, this effect might have been compensated by decreased organic burial on continental slopes due to shallower LGM sea levels[86,87]. Considering the short (~10,000 years) last deglacial[84], we assume that global PO₄ and Cd reservoirs remained constant between the Holocene and LGM. Our reconstructions (Fig. 3) are consistent with high benthic δ¹³C and low benthic Cd/Ca at numerous glacial North Atlantic mid-depth sites[23,31,46,65,88,89].

**Deep-water temperature and salinity estimates.** Deep-water temperature ($T_{deep}$) is estimated from the ice volume corrected benthic δ¹⁸O (δ¹⁸O_IVC) and the δ¹⁸O-temperature equation of Marchitto et al.[90] from $T_{deep} = 2.5 - (\delta^{18}O_{IVC} - 2.8)/0.224$, where $\delta^{18}O_{IVC} = \delta^{18}O_{benthic} - \delta_{18}O_{global\_sealevel}$. $\delta^{18}O_{global\_sealevel}$ was estimated from sea level curves[86,87] with a global δ¹⁸O_seawater−sea level scaling of 0.0085‰/m (ref. [91]). Deep-water salinity ($S_{deep}$) is calculated by: $S_{deep} = S_{core\_top} + 1.11 \times \delta^{18}O_{global\_sealevel}$, where $S_{core\_top}$ is the modern $S_{deep}$ (35.06, 34.926, and 34.893 at BOFS 17, 11, and 14 K, respectively)[2] and the term 1.11 is the scaling term for a global S–δ¹⁸O_global_sealevel relationship[29,91]. We assume ±1 °C and ±1‰ uncertainties in $T_{deep}$ and $S_{deep}$, respectively. Use of other methods to estimate $T_{deep}$ and $S_{deep}$ negligibly affects our conclusions, due to relatively weak sensitivities of [CO₃²⁻]_Norm to T and S changes (Fig. 4).

**Uncertainties and statistical analyses.** Uncertainties associated with [CO₃²⁻] and PO₄ were evaluated using a Monte-Carlo approach[92,93]. Errors associated with the chronology (x-axis) and [CO₃²⁻] and PO₄ reconstructions (y-axis) are considered during error propagation. Age errors are assumed to be ±3000 years for the three BOFS cores. Methods to calculate errors associated with individual [CO₃²⁻] and PO₄ reconstructions (y-axis) are given above. All data points were sampled and randomly 5000 times within their chronological and [CO₃²⁻] or PO₄ uncertainties and each iteration was then interpolated linearly. At each time step, the probability maximum and data distribution uncertainties of the 5000 iterations were assessed. Figure 3 shows probability maxima (bold curves) and ±95% (light gray; 2.5–97.5th percentile) probability intervals for the data distributions, including chronological and proxy uncertainties. For details, see refs. [92,93].

For a time period (e.g., Holocene) where multiple analyses are available, uncertainties are calculated following the method from ref. [94] by $2\sigma = \sqrt{[\sum_{i=1}^{n}(2\sigma_i)^2]/n}$, where $n$ is the number of reconstructions and $2\sigma_i$ is the error associated with individual reconstruction. For [CO₃²⁻] or [CO₃²⁻]_Norm offsets between the Holocene and LGM, $2\sigma = \sqrt{(2\sigma_{Holocene})^2 + (2\sigma_{LGM})^2}$, where $2\sigma_{Holocene}$ and $2\sigma_{LGM}$ are 2σ of Holocene and LGM values, respectively. Other methods (e.g., weighted mean)[95] would give similar results.

When using Eq. (11) to calculate [CO₃²⁻]_Norm, errors from various sensitivities are <1.5 μmol/kg (see Supplementary Data 8 for crosschecking). Because [CO₃²⁻]_Norm is normalized to a constant condition (i.e., no error with final T–S–P), the error in [CO₃²⁻]_Norm is largely sourced from [CO₃²⁻] reconstruction uncertainties. For surface water [CO₃²⁻]_Norm calculations, T and S errors are already included in surface [CO₃²⁻] reconstructions. For calculations associated with deep waters, [CO₃²⁻]_Norm errors are ~0.5, ~3.5, and ~0.1 μmol/kg from ±1 °C in T, ±1‰ in S, and ±50 dbar in P, respectively. Therefore, these uncertainties (already included in error calculations) are relatively less important compared to the reconstruction error of ±10 μmol/kg for deep water [CO₃²⁻].

## Data availability
The data reported in the paper are presented in Supplementary Data.

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

## Acknowledgements

This contribution is dedicated to W.S. Broecker. We thank R.F. Anderson and Daniel Sigman for the discussions. This work is supported by ARC Future Fellowship (FT140100993) and Discovery Projects (DP140101393 and DP190100894) and NSFC (41676026) to J.Y., DECRA (DE150100107) and Discovery (DP180100048) to L.M., NERC Advanced Fellowship to GLF (NE/D00876X/2), and Australian Laureate Fellowship (FL120100050) to E.J.R.

## Author contributions

J.Y. conceived the idea and wrote the paper. L.M. assisted with the model data used. Z.J./F.Z. picked the foram shells. E.J.R./Y.D. assisted with the statistics. G.L.F./J.Y. measured the boron isotopes. All authors (L.M., Z.J., D.T., G.F., E.R., N.M., J.M., Y.D., H.R., F.H., F.Z., P.C. and A.R.) contributed to improving the paper.
