## [Peer Review File · Nature Communications]

Reviewers' comments:

Reviewer #1 (Remarks to the Author):

Review of 'More efficient North Atlantic carbon pump during the Last Glacial Maximum'

This manuscript presents an interesting conundrum. On one hand, $\delta^{11}\text{B}$ measurements on planktonic foraminifera indicate elevated surface-ocean $[\text{CO}_3^{2-}]$ during the Last Glacial Maximum (LGM), consistent with the observed lower atmospheric pCO_2 . On the other hand, $\delta^{11}\text{B}$ and B/Ca measurements on benthic foraminifera indicate similar to preindustrial values for LGM $[\text{CO}_3^{2-}]$ in the intermediate-depth North Atlantic. Somehow, the vertical $[\text{CO}_3^{2-}]$ gradient must have been larger during the LGM. Arguably, the most obvious explanation would be a larger pool of regenerated organic matter, but this appears inconsistent with the Cd/Ca proxy for nutrients. Also after taking into account glacial-interglacial changes in temperature and salinity, the inferred larger LGM vertical $[\text{CO}_3^{2-}]$ remains largely unexplained.

The authors attempt to explain the large vertical $[\text{CO}_3^{2-}]$ gradient through a more efficient carbon pump during the LGM. This seems illogical to me, because the authors have already taken the soft-tissue and solubility carbon pumps into account in their calculations. If anything, the calculations by the authors tend to overestimate the impacts of the carbon pumps because of incomplete air-sea equilibration of carbon (Toggweiler et al., 2003) or incomplete nutrient utilization (Ito & Follows, 2005). In other words, the main carbon pumps cannot be stronger than what the authors already accounted for, although they can be weaker.

How to resolve this conundrum? The authors make use of many different proxies involving many different assumptions. Could one of those proxies perhaps be less accurate than the authors seem to think? There has been much debate about the range of applicability of $\delta^{11}\text{B}$ and B/Ca as pH proxies (Pagani et al., 2005; Yu et al., 2010; Rae et al., 2011) and of Cd/Ca as a proxy for nutrients (De Baar et al., 1994; Janssen et al., 2014). Possible inaccuracies in the reconstructed concentrations may be a somewhat dissatisfactory explanation, but it seems much more viable than what the authors are suggesting now. Overall, I can't support publication of this manuscript in Nature Communications, even though it raises interesting questions. In my view, the explanation of the main results is simply not compelling enough.

References

- De Baar, H.J.W., et al., *Mar. Chem.* 46, 261-281 (1994)
Ito, T., and Follows, M.J., *J. Mar. Res.* 63, 813-839 (2005)
Janssen, D.J., et al., *PNAS* 111, 6888-6893 (2014)
Pagani, M., et al., *Geochim. Cosmochim. Acta* 69, 953-961 (2005)
Rae, J.W.B., et al., *Earth Planet. Sc. Lett.* 302, 403-413 (2011)
Toggweiler, J.R., et al., *Glob. Biogeochem. Cyc.* 17, 1026 (2003)
Yu, J., et al., *Earth Planet. Sc. Lett.* 293, 114-120 (2010)

Reviewer #2 (Remarks to the Author):

This is a very interesting manuscript, a first to try and tackle changes in the Atlantic carbon pump efficiency over glacial-interglacial timescales, with implications for atmospheric CO_2 . The authors propose that the carbon pump of the North Atlantic tripled during the LGM compared with the Holocene.

To come to this conclusion they use estimates of estimate air-sea CO_2 exchange signals using carbonate ion reconstructions for surface waters in a core from the Caribbean (not Gulf stream),

and multiple cores from ~intermediate depth in the North Atlantic, and supplement these with nutrient reconstructions of the same North Atlantic cores.

I think this approach is very novel, although several major generalizations are made with regards to the interpretations of the various proxies involved, something I am sure the authors are aware of.

One generalization involves the mode of deep water formation in the North Atlantic, which was supposed to be considerable different and in a different location during the LGM compared with today. Potential effects of sea-ice also are ignored.

Furthermore, can ODP 999 from the Caribbean really be considered representative for the Gulf Stream over the entire interval? For example during early deglaciation, Leduc et al. (2007) suggest that orogenic blocking of the Andes during southward ITCZ movement caused this fresh water to be returned to the Atlantic Ocean, via the Amazon basin drainage, which led to lowered salinity of low-latitude currents in the Atlantic Ocean, such as the North Brazil current, the Guyana current and the Caribbean current feeding the Gulf Stream.

Barker and Elderfield (2000) propose carbonate ion in the North Atlantic surface waters was up from Holocene values of about 200 $\mu\text{mol/kg}$ to glacial values of 260 $\mu\text{mol/kg}$. This is a lot smaller compared with the increase at the Caribbean site.

Cd/Ca relationships discussed in the paper are based on very old papers of Ed Boyle. Marchitto and Broecker (2006, G-cubed) show that there are issues with using Cd/Ca in the intermediate North Atlantic for reconstructing phosphate. Real time measurements put phosphate concentrations at these locations below 1.2 $\mu\text{mol/kg}$. Reconstructed values by the authors are all equal to or in excess of 1.3 $\mu\text{mol/kg}$, which are found much further South today.

Marchitto and Broecker (2006) suggest that Cd should be close to 0.2 nmol/kg . At the core locations in this study these seem to be values achieved during the glacial, not during the Holocene.

Actual phosphate values at the various locations should be shown in Figure 3, and the authors should discuss in the manuscript why they think their values can be considered robust given this discrepancy.

Was Cd/Ca measured at ODP 999, as there is a value of 0 in Figure 5...

In addition, it would be good to discuss about why Cd/Ca in intermediate bottom waters were even lower during the LGM compared with today where they are already one of the most nutrient depleted intermediate water masses.

Today the Caribbean may be considered as a slight source of atmospheric CO_2 . Is there any way the data can tell you whether it became less or more of a source of CO_2 during the glacial, and what would the implications be for the carbonate ion gradient interpretation?

Finally, surface water $\delta^{18}\text{O}$ of the Caribbean became heavier during the LGM (+0.4 per mil after ice volume correction); but in the North Atlantic they decrease (Thornalley et al., 2010). If this is in relation to sea-ice/brine rejection, can nutrient versus $\delta^{18}\text{O}_{\text{sw}}$ relationships in areas with sea-ice really be regarded as strong as ice-free open ocean?

Small comment/question: boron isotope measurements are from Foster (2008), whereas in the manuscript it seems to be described as new measurements?

Reviewer #3 (Remarks to the Author):

I find this paper fascinating. After reading through the main text and the supplement, I don't see critical flaws in this study. Therefore, I recommend its publication in Nature Communications after minor revisions. As I see it, this study should be of broad interest to the paleoclimate communities as well as the general readers.

It has long been a grand challenge for quantifying how much additional carbon was captured by the ocean during glacial periods. Using Last Glacial Maximum (LGM) as a classic example, the authors developed a novel and applicable way to tackle this problem. They utilized multiple proxies (e.g. boron isotopes, Mg/Ca, B/Ca, Cr/Ca ratios) to generate a powerful tracer, called [CO₃²⁻]_{as}, which could be used to distinguish the air-sea exchange effect from the biological and other physical effects on changing the seawater carbonate system. By comparing the surface ocean carbonate chemistry in the Caribbean with deep ocean carbonate chemistry in the North Atlantic both in Holocene and LGM, the change of carbon sink of North Atlantic Ocean in the LGM relative to the Holocene is determined and the previously underappreciated contribution of North Atlantic Ocean is indicated. As the title suggested, the conclusion is that we have more efficient North Atlantic carbon pump during the LGM.

This paper also reads well. The experimental and sample details are given (mostly in the SI), the comprehensive numerical method descriptions are listed, and the conclusion looks robust to me. There are many definitions of new terms in this paper, such as [DIC]_{as}, [CO₃²⁻]_{as}, [CO₃]EXP, [CO₃]EXP-in-situ. These parameters are critical for understanding the essence of this research. I find the authors give a clear explanation for these conceptions and make them accessible to the readers. Their calculation procedures are provided in the method part and the supplementary materials, which I think is really crucial for the readers to understand how this study rigorously evaluates the North Atlantic carbon sequestration during the Last Glacial Maximum.

It is always a hassle when dealing with so many proxies because the error is difficult to quantify. To handle this problem, the authors adopt all three approaches—quadratic addition of individual errors, Monte Carlo resampling and creating several scenarios. In this way, errors are fully propagated and various situations are explored.

My major concern is what is the relative contribution of North Atlantic Ocean in sinking CO₂ compared with the Southern Ocean. We know from this study that North Atlantic CO₂ pump efficiency during the LGM was enhanced by a factor of ~2.7 relative to the Holocene. I am wondering if there is any way to quantitatively (or even qualitatively) constrain the absolute carbon sink fluxes from the NA. This is kind of important to know better the role NA played in LGM. If the carbon sink flux turns out to be small, then even if the efficiency of NA was greatly increased during LGM, it might still play a minor role. Some discussion of this aspect should be given in the text.

Below are several minor comments:

Line 50 and Line 52: change n-dash to m-dash

Line 74-76: I think strong cooling will decrease the Henry constant, and thus there will be more CO₂ in the ocean relative to the atmosphere. It might be not proper to say "strong cooling causes surface-water pCO₂ to be lower than atmospheric pCO₂". The high nutrient utilization does causes surface-water pCO₂ to be lower than atmospheric pCO₂. Need to rephrase.

Line 94-96: Does this sentence indicate efficient biological pump is caused by "strong cooling of low-PO₄ northward-flowing Gulf Stream waters and limited nutrient supply from the subsurface"? Need to rewrite the sentence.

Line 154: dissociated constant should be "dissociation constant". Similar to Line 110.

Line 161: delete the comma after “,”.

Line 245-246: change “for some of which” to “for which some”?

Dear Reviewers,

We have revised our manuscript substantially based on the valuable review comments received. The constructive comments have improved our manuscript significantly. In addition to changes to address the points raised, we emphasize the pragmatic recipe in the main text since it is easier to follow and apply. The second method, which involves frequent use of the CO₂sys software, has now been moved to the SI. Some redundant supplementary figures (mainly related to the CO₂sys sensitivity test) have also been removed from the SI. We hope you find the revision more streamlined.

Below, we list our point-by-point responses (blue) to your comments (black). The main text has been revised accordingly (blue).

===

Reviewer #1 (Remarks to the Author):

Review of 'More efficient North Atlantic carbon pump during the Last Glacial Maximum'

This manuscript presents an interesting conundrum. On one hand, $\delta^{11}\text{B}$ measurements on planktonic foraminifera indicate elevated surface-ocean $[\text{CO}_3^{2-}]$ during the Last Glacial Maximum (LGM), consistent with the observed lower atmospheric pCO_2 . On the other hand, $\delta^{11}\text{B}$ and B/Ca measurements on benthic foraminifera indicate similar to preindustrial values for LGM $[\text{CO}_3^{2-}]$ in the intermediate-depth North Atlantic. Somehow, the vertical $[\text{CO}_3^{2-}]$ gradient must have been larger during the LGM. Arguably, the most obvious explanation would be a larger pool of regenerated organic matter, but this appears inconsistent with the Cd/Ca proxy for nutrients. Also after taking into account glacial-interglacial changes in temperature and salinity, the inferred larger LGM vertical $[\text{CO}_3^{2-}]$ remains largely unexplained.

This is a good summary of our study.

The authors attempt to explain the large vertical $[\text{CO}_3^{2-}]$ gradient through a more efficient carbon pump during the LGM. This seems illogical to me, because the authors have already taken the soft-tissue and solubility carbon pumps into account in their calculations. If anything, the calculations by the authors tend to **overestimate** the impacts of the carbon pumps because of incomplete air-sea equilibration of carbon (Toggweiler et al., 2003) or incomplete nutrient utilization (Ito & Follows, 2005). In other words, the main carbon pumps cannot be stronger than what the authors already accounted for, although they can be weaker.

There may be a misunderstanding here about our correction method to account for influences from the solubility and biological pumps.

Our approach is to isolate the air-sea exchange component CO_2 in the ocean. Corrections are necessary for each of the Holocene and LGM. To be simple, let us consider the Holocene. If our method tends to “overestimate” carbon pump impacts as described by the reviewer, it would be impossible to explain the vertical $[\text{CO}_3^{2-}]$ gradient even for the Holocene where our reconstructions are consistent with estimates using the GLODAP dataset (Figs. 2, 5).

We think that the misunderstanding surrounds what is corrected by our method. Below, we explain details about our correction method.

Regarding the solubility pump, it affects $[\text{CO}_3^{2-}]$ in two ways:

- (i) any T/S change would change CO_2 system dissociation constants and thus $[\text{CO}_3^{2-}]$, and
- (ii) when at the surface, T/S variations would change air-sea $p\text{CO}_2$ gradients, alter air-sea exchange component CO_2 in seawater, and thereby affect $[\text{CO}_3^{2-}]$.

Regarding the biological pump, its effects on $[\text{CO}_3^{2-}]$ include two parts:

- (i) changes in nutrient utilization and respiration redistribute DIC and ALK within the ocean and hence $[\text{CO}_3^{2-}]$ (see changes from Fig. 1a to Fig. 1b), and
- (ii) biological processes would change air-sea $p\text{CO}_2$ gradients, alter air-sea CO_2 exchanges (see changes from Fig. 1b to Fig. 1c), and thereby affect $[\text{CO}_3^{2-}]$.

For both pumps, our corrections account for influences only from (i), assuming zero net air-sea CO_2 change from (ii).

Therefore, our calculations represent the minimum impact (i.e., not to overestimate any effect) of the solubility and biological pumps on $[\text{CO}_3^{2-}]$, because it assumes no air-sea CO_2 exchange. This is critical to interpret the $[\text{CO}_3^{2-}]$ gradients presented in our manuscript.

After the above corrections, any $[\text{CO}_3^{2-}]$ gradient would reflect air-sea CO_2 exchange. The more air-sea CO_2 uptake, the larger the $[\text{CO}_3^{2-}]$ gradient. In other words, after our corrections, more CO_2 invasion would cause greater vertical $[\text{CO}_3^{2-}]$ gradients.

Given the Reviewer's comment above, we have further clarified our correction method in the main text wherever appropriate (lines 108-109, 169-178, 188, 720). In particular, we make use of well-defined sensitivities (Fig. 4; Methods) and a plot of $[\text{CO}_3^{2-}]_{\text{Norm}}$ versus PO_4 (Fig. 5) to demonstrate the corrections associated with the solubility and biological pumps, because they are more straightforward to understand.

We hope that our clarification resolves the reviewer's comment.

How to resolve this conundrum? The authors make use of many different proxies involving many different assumptions. Could one of those proxies perhaps be less accurate than the authors seem to think? There has been much debate about the range of applicability of $d_{11}\text{B}$ and B/Ca as pH proxies (Pagani et al., 2005; Yu et al., 2010; Rae et al., 2011) and of Cd/Ca as a proxy for nutrients (De Baar et al., 1994; Janssen et al., 2014). Possible inaccuracies in the reconstructed concentrations may be a somewhat dissatisfactory explanation, but it seems much more viable than what the authors are suggesting now. Overall, I can't support publication of this manuscript in Nature Communications, even though it raises interesting questions. In my view, the explanation of the main results is simply not compelling enough.

References

- De Baar, H.J.W., et al., Mar. Chem. 46, 261-281 (1994)
Ito, T., and Follows, M.J., J. Mar. Res. 63, 813-839 (2005)
Janssen, D.J., et al., PNAS 111, 6888-6893 (2014)
Pagani, M., et al., Geochim. Cosmochim. Acta 69, 953-961 (2005)

Rae, J.W.B., et al., Earth Planet. Sc. Lett. 302, 403-413 (2011)
Toggweiler, J.R., et al., Glob. Biogeochem. Cyc. 17, 1026 (2003)
Yu, J., et al., Earth Planet. Sc. Lett. 293, 114-120 (2010)

Our conclusions mainly rely on three proxies: (i) planktonic $\delta^{11}\text{B}$ for surface water $[\text{CO}_3^{2-}]$, (ii) benthic $\delta^{11}\text{B}$ and B/Ca for deep water $[\text{CO}_3^{2-}]$, and (iii) benthic Cd/Ca for deep water PO_4 .

The reliability of our $[\text{CO}_3^{2-}]$ reconstructions is supported by the following evidence:

- The criticism of $\delta^{11}\text{B}$ by Pagani et al. (2005) was published a long time ago. Since then, a new analytical method was developed using a Neptune⁺ instrument in 2008 (ref. ¹), and the $\delta^{11}\text{B}$ proxy has been re-calibrated using cultured and core-top samples^{2,3}.
- Yu et al. (2010) and Rae et al. (2011) are by the authors (Yu and Foster) of this manuscript, and provide core-top and down-core data to support the reliability of B/Ca and $\delta^{11}\text{B}$ for deep water $[\text{CO}_3^{2-}]$ reconstructions.
- After publication of Yu et al. (2010), even the late Mark Pagani was convinced about the faithfulness of the $\delta^{11}\text{B}$ and B/Ca proxies. See the attached comment from Mark Pagani at the end of this rebuttal.
- Our new benthic $\delta^{11}\text{B}$ (using the Neptune⁺ method)¹ and B/Ca data yield consistent $[\text{CO}_3^{2-}]$ results for the Holocene and LGM (Fig. 3).

Regarding the benthic Cd/Ca proxy, there is some debate. However, all published studies unanimously suggest lower PO_4 for the Glacial North Intermediate Water (GNAIW), which is consistent with its much elevated benthic $\delta^{13}\text{C}$ ($\sim 1.5\text{‰}$)⁴⁻¹⁰. To have no increase in the North Atlantic CO_2 uptake efficiency would require a higher PO_4 for GNAIW (Fig. R1; next page), which contradicts proxy data.

Instead of the proxies, we suspect that the real issue with the comments from Reviewer 1 surrounds a misunderstanding of our correction method. With the clarifications provided here, we hope that the reviewer can now agree with our conclusions.

Additionally, temperature, salinity, and pressure are estimated using established approaches. Their effects on seawater $[\text{CO}_3^{2-}]$ are based on well-defined sensitivities (Fig. 4). All uncertainties are fully propagated as described in the SI (see appraisal of our error estimates from Reviewer 3).

Finally, we thank Reviewer 1 for the valuable comments, which we have used to significantly improve the description of our correction method.

Fig. R1. Influence of GNAIW PO_4 on ODP 999–BOFS $[CO_3^{2-}]_{Norm}$ offset. Holocene data are as shown in Fig. 5b. For the LGM, we *assume* that deep-water PO_4 at BOFS sites can be at any value ($[CO_3^{2-}]$ values at these sites are from B/Ca- $\delta^{11}B$ reconstructions). To reach comparable ODP 999–BOFS $[CO_3^{2-}]_{Norm}$ offsets between the Holocene and LGM, it would require LGM deep-water PO_4 at BOFS sites to be $\sim 0.4 \mu\text{mol/kg}$ higher than the Holocene, which is not supported by proxy data. If necessary, please see “A pragmatic recipe to estimate $[CO_3^{2-}]_{as}$ change” section in the main text for use of the $[CO_3^{2-}]_{Norm}$ - PO_4 plot.

Reviewer #2 (Remarks to the Author):

This is a **very interesting** manuscript, **a first** to try and tackle changes in the Atlantic carbon pump efficiency over glacial-interglacial timescales, with implications for atmospheric CO_2 . The authors propose that the carbon pump of the North Atlantic tripled during the LGM compared with the Holocene. To come to this conclusion they use estimates of estimate air-sea CO_2 exchange signals using carbonate ion reconstructions for surface waters in a core from the Caribbean (not Gulf stream), and multiple cores from \sim intermediate depth in the North Atlantic, and supplement these with nutrient reconstructions of the same North Atlantic cores.

I think this approach is **very novel**, although several major generalizations are made with regards to the interpretations of the various proxies involved, something I am sure the authors are aware of. One generalization involves the mode of deep water formation in the North Atlantic, which was supposed to be considerable different and in a different location during the LGM compared with today. Potential effects of sea-ice also are ignored.

We thank the reviewer for commenting on the novelty of our work. Below we address specific points.

Furthermore, can ODP 999 from the Caribbean really be considered representative for the Gulf Stream over the entire interval? For example during early **deglaciation**, Leduc et al. (2007) suggest that orogenic blocking of the Andes during southward ITCZ movement caused this fresh water to be returned to the Atlantic Ocean, via the Amazon basin drainage, which led to lowered salinity of low-latitude currents in the Atlantic Ocean, such as the North Brazil current, the Guyana current and the Caribbean current feeding the Gulf Stream.

Thank you for this insightful comment on the use of ODP 999 for Gulf Stream reconstructions.

We use ODP 999 to constrain Gulf Stream chemistry because (i) this core has published $\delta^{11}\text{B}$ ready for use¹, and (ii) most North Atlantic subtropical gyre water circulates through the Caribbean Sea before being transported to the subpolar North Atlantic via the Gulf Stream¹¹.

As a first attempt with our approach, we think it is reasonable to use ODP 999 to constrain the first-order carbonate chemistry of warm surface source water changes. We hope the reviewer agrees.

We agree with the reviewer's point about possible complications during the last deglacial. Therefore, we have removed the deglacial calculations. Fig. 3 has been revised accordingly.

In the revision, we now focus on two time intervals: the Holocene and LGM.

Following the reviewer's advice, we have further investigated whether ODP 999 can be used to represent the first-order carbonate chemistry changes for the Holocene and LGM. To do so, we have tried to reconstruct surface water $[\text{CO}_3^{2-}]$ for four additional cores from the wider tropical western Atlantic. For these cores, we assume quasi-equilibrium in $p\text{CO}_2$ between surface waters and the atmosphere. We believe that this is a reasonable assumption because subtropical surface waters cycle multiple times through the North Atlantic gyre (refs ^{1,2}). As can be seen from Fig. S6, ODP 999 and the other four sites show very similar $[\text{CO}_3^{2-}]_{\text{Norm}}$ during the Holocene and LGM. We therefore believe that it is reasonable to use ODP 999 to represent the first-order Gulf Stream carbonate chemistry for the two time intervals of interest here.

We have explained our reasoning above in the main text (lines 131-140) and SI (Section 3.3).

Fig. S6. Similar [CO₃²⁻]_{Norm} at ODP 999 and other sites from the broader subtropical western North Atlantic. **a**, Map showing core locations. **b**, [CO₃²⁻] for the Holocene and LGM. **c**, as **b** but for [CO₃²⁻]_{Norm}. [CO₃²⁻]_{Norm} for ODP 999 is calculated using δ¹¹B derived pH and ALK, while for other cores [CO₃²⁻]_{Norm} is based on the assumption of surface-atmosphere pCO₂ equilibrium within ±15 ppm. Note that [CO₃²⁻]_{Norm} (not [CO₃²⁻]) should be used for air-sea CO₂ exchange calculations (see Fig. 5).

Barker and Elderfield (2000) propose carbonate ion in the North Atlantic surface waters was up from Holocene values of about 200 μmol/kg to glacial values of 260 μmol/kg. This is a lot smaller compared with the increase at the Caribbean site.

We find very similar Holocene-LGM [CO₃²⁻] changes (~60 μmol/kg) between the two studies. Please refer to Fig. 3a for our ODP 999 [CO₃²⁻] reconstructions.

Maybe, the reviewer meant the absolute [CO₃²⁻] contrast between ODP 999 and Barker's NEAP 8K? If so, please bear in mind that absolute values are not directly comparable due to different T-S-PO₄ conditions between sites (see Fig. 4).

Cd/Ca relationships discussed in the paper are based on very old papers of Ed Boyle. Marchitto and Broecker (2006, G-cubed) show that there are issues with using Cd/Ca in the intermediate North Atlantic for reconstructing phosphate. Real time measurements put phosphate concentrations at these locations below 1.2 μmol/kg. Reconstructed values by the authors are all equal to or in excess of 1.3 μmol/kg, which are found much further South today. Marchitto and Broecker (2006) suggest that Cd should be close to 0.2 nmol/kg. At the core locations in this study these seem to be values achieved during the glacial, not during the Holocene.

This is a great point! Thanks again for this insightful comment.

In the previous version, we simply used previous D_{Cd} values from Bertram et al. (1995)¹² to reconstruct deep water Cd and PO₄ values. As correctly pointed out by the reviewer, our calculations give too high (by ~25%) Cd values for the Holocene. This offset may suggest higher D_{Cd} values for the North Atlantic Ocean, which has been acknowledged recently¹³. Accordingly, we have adjusted D_{Cd} values so that new derived Holocene Cd values match the preindustrial value of ~0.25 nmol/kg. All associated calculations have been updated accordingly (see Section 4.2 in SI).

Fig. S7. Seawater Cd and PO₄ from the North Atlantic. a, Bathymetric Cd profiles for the North Atlantic¹⁴.

Actual phosphate values at the various locations should be shown in Figure 3, and the authors should discuss in the manuscript why they think their values can be considered robust given this discrepancy.

Thank you. We have updated Fig. 3 to show PO₄ values.

To justify the robustness of our down-core nutrient estimates, we have compared our reconstructions with those based on *H. elegans*. Compared to *Cibicidoides* (used in this study), D_{Cd} into *H. elegans* is far less variable¹⁵. As can be seen from Fig. S8 (next page), for cores bathed in water masses with similar benthic $\delta^{13}C$, our Cd reconstructions match favourably with those based on *H. elegans*¹⁰. This supports the reliability of our reconstructions.

Using updated PO₄ reconstructions, we find glacial North Atlantic carbon uptake efficiency was doubled compared to the Holocene. **Note:** our conclusion of a more efficient LGM carbon pump remains unchanged, as long as LGM PO₄ was lower than in the Holocene even without knowing the exact value of LGM PO₄. See Fig. R1 above for details.

Fig. S8. Comparison of Cd from BOFS 17K and other cores. **a**, Benthic $\delta^{13}\text{C}$. Similar $\delta^{13}\text{C}$ values suggest that cores were possibly bathed in the same water mass, namely GNAIW, during the LGM. **b**, Deep water Cd reconstructed using *Cibicidoides* from BOFS 17K (58°N, 16.5°W, 1150 m), and *H. elegans* from 100GGC (26°N, 78°W, 1057 m) and 103GGC (26°N, 78°W, 965 m)¹⁰. The three cores are from similar water depths with comparable benthic $\delta^{13}\text{C}$. Similar Cd from three cores supports the validity of BOS 17K Cd reconstructions.

Was Cd/Ca measured at ODP 999, as there is a value of 0 in Figure 5...

This is assumed due to its oligotrophic setting (lines 147-148). Any PO_4 increase at ODP 999 during the LGM would strengthen our conclusion (see sensitivity test in Fig. S17b).

Fig. S17. b, Surface-water PO_4 effect. Any increase in PO_4 at ODP 999 would raise the ODP 999-BOFS $[\text{CO}_3^{2-}]_{\text{as}}$ gradient during the LGM. This would suggest greater glacial North Atlantic carbon sequestration, strengthening our conclusions.

In addition, it would be good to discuss about why Cd/Ca in intermediate bottom waters were even lower during the LGM compared with today where they are already one of the most nutrient depleted intermediate water masses.

Despite the well accepted view of decreased GNAIW nutrient levels, the mechanisms causing this change are not fully clear. Possible reasons at least include: (i) lower preformed nutrient (and hence a stronger biological pump), (ii) reduced respiration related to faster ventilation, and (iii) reduced nutrient supply from glacial AAIW. Further work is needed to distinguish their relative roles in decreasing PO_4 of GNAIW.

Based on the reviewer's advice, we have now briefly discussed reasons for the lower Cd in GNAIW (lines 266-268).

Today the Caribbean may be considered as a slight source of atmospheric CO_2 . Is there any way the data can tell you whether it became less or more of a source of CO_2 during the glacial, and what would the implications be for the carbonate ion gradient interpretation?

The reviewer is correct in that Caribbean today (preindustrial) is indeed a weak source of CO_2 to the atmosphere. This may also have been true for the Holocene and LGM when surface water pCO_2 was higher than the atmosphere by ~ 20 ppm (Fig. R2; next page). Everything else being equal, comparable pCO_2 gradients would suggest similar CO_2 outgassing from $\sim 10^\circ N$ (Caribbean) to $\sim 35^\circ N$ (latitudes north of which we define as the North Atlantic) for the Holocene and LGM. Thus, our conclusion remains unaffected.

Given large pCO_2 reconstruction uncertainties ($\sim \pm 27$ ppm) and other factors (e.g., wind, air-sea contact time) that affect air-sea CO_2 exchange fluxes, we think it is more meaningful to consider temporal magnitudes of changes in other carbonate chemistry variables (e.g., DIC_{as} , $[CO_3^{2-}]_{Norm}$) as we have done here.

Preindustrial Caribbean hydrographic sites show similar DIC_{as} values as sites located close to the Gulf Stream (Fig. S3). For the Holocene and LGM, ODP 999 also shows similar $[CO_3^{2-}]_{Norm}$ values to the other four sites from the western tropical Atlantic sites (Fig. S6). Because we have assumed air-sea CO_2 equilibrium and hence minimal net air-sea CO_2 flux at these four sites, comparable $[CO_3^{2-}]_{Norm}$ values suggest that air-sea exchange CO_2 signals do not change much in the tropical Atlantic region.

Therefore, any CO_2 outgassing in Caribbean would have an insignificant impact on our conclusions, because waters at low latitudes ($< \sim 35^\circ N$) are more or less at equilibrium with atmospheric CO_2 ^{16,17} (lines 115-116).

Fig. R2. Surface water $p\text{CO}_2$ vs. $\Delta p\text{CO}_2$ for ODP 999, calculated after ^{1,2}. $\Delta p\text{CO}_2$ is the $p\text{CO}_2$ difference between surface waters and the atmosphere.

Finally, surface water $\delta^{18}\text{O}$ of the Caribbean became heavier **during the LGM** (+0.4 per mil after ice volume correction); but in the North Atlantic they decrease (Thornalley et al., 2010). If this is in relation to sea-ice/brine rejection, can nutrient versus $\delta^{18}\text{O}_{\text{sw}}$ relationships in areas with sea-ice really be regarded as strong as ice-free open ocean?

We don't see very different $\delta^{18}\text{O}_{\text{sw-ivc}}$ changes between Caribbean and the polar North Atlantic surface waters. Thornalley et al. (2010)¹⁸ show no data for the LGM (pre ~18 ka) because *G. bulloides* and *G. inflata* are so scarce (according to David Thornalley, a co-author of this study). The lighter *G. bulloides* and *G. inflata* $\delta^{18}\text{O}_{\text{sw-ivc}}$ during HS1 were argued to be caused by the input of Laurentide ice sheet meltwater.

The only LGM data shown are from *N. pachyderma* (s) and display heavier LGM $\delta^{18}\text{O}_{\text{sw-ivc}}$, which is consistent with the Caribbean with apparently similar magnitude of changes of ~+0.4‰ and ~+0.5‰, respectively¹⁹ (Fig. R3; next page). Additionally, Benway et al. (2010)²⁰ also report heavier *N. pachyderma* (s) $\delta^{18}\text{O}_{\text{sw-ivc}}$ of about +0.5‰ heavier in LGM in subpolar northeastern Atlantic (Fig. R3).

Any change with deep water production mode and sea-ice formation would have little impact on our calculations, because our interest lies in net carbonate chemistry changes between surface (ODP 999) and deep waters (BOFS cores) (lines 121-123). However, we do appreciate the reviewer's view that knowledge of past polar surface hydrography would improve our understanding of mechanisms causing past carbon cycle changes. To remain focused, we leave this topic for future studies.

Fig. R3. Published $\delta^{18}\text{O}_{\text{sw-ivc}}$ for the polar North Atlantic. Left, data from Thornalley et al. (2011)¹⁹. Right, data from Benway et al. (2010)²⁰.

Small comment/question: boron isotope measurements are from Foster (2008), whereas in the manuscript it seems to be described as new measurements?

ODP 999 *G. ruber* $\delta^{11}\text{B}$ data are from Foster (2008)¹, while benthic $\delta^{11}\text{B}$ data are from this study. We have clarified this in the main text. Data sources are also provided in the supplementary tables.

Finally, we appreciate the reviewer for the careful review, especially the insightful advice on deep water Cd reconstructions. With your inputs, we believe our conclusion has been significantly strengthened. Thanks!

Reviewer #3 (Remarks to the Author):

I find this paper **fascinating**. After reading through the main text and the supplement, I don't see critical flaws in this study. Therefore, I recommend its publication in Nature Communications after minor revisions. As I see it, this study should be of broad interest to the paleoclimate communities as well as the general readers.

It has long been a grand challenge for quantifying how much additional carbon was captured by the ocean during glacial periods. Using Last Glacial Maximum (LGM) as a classic example, the authors developed a **novel and applicable way** to tackle this problem. They utilized multiple proxies (e.g. boron isotopes, Mg/Ca, B/Ca, Cr/Ca ratios) to generate a powerful tracer, called [CO32-]as, which could be used to distinguish the air-sea exchange effect from the biological and other physical effects on changing the seawater carbonate system. By comparing the surface ocean carbonate chemistry in the Caribbean with deep ocean carbonate chemistry in the North Atlantic both in Holocene and LGM, the change of carbon sink of North Atlantic Ocean in the LGM relative to the Holocene is determined and the previously underappreciated contribution of North Atlantic Ocean is indicated. As the title suggested, the conclusion is that we have more efficient North Atlantic carbon pump during the LGM.

This paper also **reads well**. The experimental and sample details are given (mostly in the SI), the comprehensive numerical method descriptions are listed, and the conclusion looks robust to me. There are many definitions of new terms in this paper, such as [DIC]_{as}, [CO₃₂]_{as}, [CO₃]EXP, [CO₃]EXP–in-situ. These parameters are critical for understanding the essence of this research. I find the authors give a clear explanation for these conceptions and make them accessible to the readers. Their calculation procedures are provided in the method part and the supplementary materials, which I think is really crucial for the readers to understand how this study rigorously evaluates the North Atlantic carbon sequestration during the Last Glacial Maximum.

It is always a hassle when dealing with so many proxies because the error is difficult to quantify. To handle this problem, the authors adopt all three approaches —quadratic addition of individual errors, Monte Carlo resampling and creating several scenarios. In this way, errors are fully propagated and various situations are explored.

We thank the reviewer very much for the appraisal of our work. The above is a fantastic summary of our work.

My major concern is what is the relative contribution of North Atlantic Ocean in sinking CO₂ compared with the Southern Ocean. We know from this study that North Atlantic CO₂ pump efficiency during the LGM was enhanced by a factor of ~2.7 relative to the Holocene. I am wondering if there is any way to quantitatively (**or even qualitatively**) constrain the absolute carbon sink fluxes from the NA. This is kind of important to know better the role NA played in LGM. If the carbon sink flux turns out to be small, then even if the efficiency of NA was greatly increased during LGM, it might still play a minor role. Some discussion of this aspect should be given in the text.

We thank the reviewer for this great suggestion. We have added a new section in the main text to quantify CO₂ uptake change between the Holocene and LGM. Calculation details are given in Methods. We also acknowledge uncertainties associated with our calculation in the main text (lines 230-254).

Our best estimate is an extra ~100 PgC carbon uptake by the LGM North Atlantic compared to the Holocene (Fig. 6; next page). We have discussed the role of the North Atlantic in relation to the Southern Ocean in the context of the global carbon budget change (lines 274-285, 290-292).

Figure 6 | North Atlantic CO₂ budget. The LGM-Holocene extra carbon uptake is based on Holocene-to-LGM DIC_{as} increase of 91 μmol/kg. The large red square represents our best estimate of ~100 PgC, assuming that NADW and GNAIW occupied ~50% and ~30% of the global deep ocean (>1 km), respectively²¹⁻²⁴. See Methods for calculation details.

Below are several minor comments:

Line 50 and Line 52: change n-dash to m-dash

This has been changed.

Line 74-76: I think strong cooling will decrease the Henry constant, and thus there will be more CO₂ in the ocean relative to the atmosphere. It might be not proper to say “strong cooling causes surface-water pCO₂ to be lower than atmospheric pCO₂”. The high nutrient utilization does causes surface-water pCO₂ to be lower than atmospheric pCO₂. Need to rephrase.

We have removed “along strong cooling” to avoid confusion.

Line 94-96: Does this sentence indicate efficient biological pump is caused by “strong cooling of low-PO₄ northward-flowing Gulf Stream waters and limited nutrient supply from the subsurface”? Need to rewrite the sentence.

Good point. It has been reworded to read (lines 94-96):

North Atlantic CO₂ absorption is driven by (i) an efficient solubility pump due to strong cooling of northward-flowing Gulf Stream waters and (ii) a strong biological pump associated with high nutrient utilization²⁵⁻²⁷.

Line 154: dissociated constant should be “dissociation constant”. Similar to Line 110.

Line 161: delete the comma after “,”.

Line 245-246: change “for some of which” to “for which some”?

All these comments have been revised. Thank you for these careful comments.

We thank the Reviewer very much for the valuable advice to quantify the total carbon budget change between the Holocene and LGM. We believe that the new calculations, even considering potential sizable uncertainties, provide a useful context to assess the North Atlantic’s roles in the past global carbon cycle. Much appreciated!

References used in rebuttal:

1. Foster GL. Seawater pH, pCO₂ and [CO₃²⁻] variations in the Caribbean Sea over the last 130 kyr; a boron isotope and B/Ca study of planktic foraminifera. *Earth Planet Sci Lett* 2008, **271**(1-4): 254-266. doi: 210.1016/j.epsl.2008.1004.1015.
2. Henehan MJ, Rae J, Foster GL, Erez J, Prentice KC, Kucera M, *et al.* Calibration of the boron isotope proxy in the planktonic foraminifera *Globigerinoides ruber* for use in palaeo-CO₂ reconstruction. *Earth Planet Sci Lett* 2013, **364**: 111-122.
3. Rae JWB, Foster GL, Schmidt DN, Elliott T. Boron isotopes and B/Ca in benthic foraminifera: Proxies for the deep ocean carbonate system. *Earth Planet Sci Lett* 2011, **302**(3-4): 403-413.
4. Boyle EA. Cadmium and δ¹³C paleochemical ocean distributions during the stage-2 glacial maximum. *Annu Rev Earth Planet Sci* 1992, **20**: 245-287.
5. Boyle E. The role of vertical chemical fractionation in controlling late Quaternary atmospheric carbon dioxide. *J Geophys Res* 1988, **93**(C12): 15701-15714.
6. Boyle EA, Keigwin L. North Atlantic thermohaline circulation during the past 20,000 years linked to high-latitude surface temperature. *Nature* 1987, **330**(6143): 35-40.
7. Marchitto T, Broecker W. Deep water mass geometry in the glacial Atlantic Ocean: A review of constraints from the paleonutrient proxy Cd/Ca. *Geochem Geophys Geosyst* 2006, **7**(12): doi:10.1029/2006GC001323.
8. Marchitto TM, Curry WB, Oppo DW. Millennial-scale changes in North Atlantic circulation since the last glaciation. *Nature* 1998, **393**(6685): 557-561.

9. Lynch-Stieglitz J, Adkins JF, Curry WB, Dokken T, Hall IR, Herguera JC, *et al.* Atlantic meridional overturning circulation during the Last Glacial Maximum. *Science* 2007, **316**(5821): 66-69.
10. Came RE, Oppo DW, Curry WB, Lynch-Stieglitz J. Deglacial variability in the surface return flow of the Atlantic meridional overturning circulation. *Paleoceanogr* 2008, **23**(1): Artn Pa1217.
11. Johns WE, Townsend TL, Fratantoni DM, Wilson WD. On the Atlantic inflow to the Caribbean Sea. *Deep Sea Research Part I: Oceanographic Research Papers* 2002, **49**(2): 211-243.
12. Bertram CJ, Elderfield H, Shackleton NJ, Macdonald JA. Cadmium/calcium and carbon-isotope reconstructions of the glacial northeast Atlantic Ocean. *Paleoceanogr* 1995, **10**(3): 563-578.
13. Oppo D, Gebbie G, Huang KF, Curry W, Marchitto T, Pietro KR. Data Constraints on Glacial Atlantic Water Mass Geometry and Properties. *Paleoceanography and Paleoclimatology* 2018, **33**(9): 1013-1034.
14. Schlitzer R, Anderson RF, Dodas EM, Lohan M, Geibert W, Tagliabue A, *et al.* The GEOTRACES Intermediate Data Product 2017. *Chem Geol* 2018, **493**: 210-223.
15. Boyle EA, Labeyrie L, Duplessy JC. Calcitic foraminiferal data confirmed by cadmium in aragonitic *Hoeglundina* - application to the Last Glacial Maximum in the northern Indian Ocean. *Paleoceanogr* 1995, **10**(5): 881-900.
16. Takahashi T, Sutherland SC, Wanninkhof R, Sweeney C, Feely RA, Chipman DW, *et al.* Climatological mean and decadal change in surface ocean pCO₂, and net sea-air CO₂ flux over the global oceans. *Deep-Sea Res Part II* 2009, **56**(8-10): 554-577.
17. Key RM, Kozyr A, Sabine CL, Lee K, Wanninkhof R, Bullister JL, *et al.* A global ocean carbon climatology: Results from Global Data Analysis Project (GLODAP). *Glob Biogeochem Cycle* 2004, **18**(4): doi: 10.1029/2004GB002247.
18. Thornalley DJR, McCave IN, Elderfield H. Freshwater input and abrupt deglacial climate change in the North Atlantic. *Paleoceanogr* 2010, **25**: PA1201, doi: 10.1029/2009PA001772.
19. Thornalley DJR, Elderfield H, McCave IN. Reconstructing deglacial North Atlantic surface hydrography and its link to the Atlantic overturning circulation. *Global and Planetary Change* 2011: doi:10.1016/j.gloplacha.2010.1006.1003.
20. Benway HM, McManus JF, Oppo DW, Cullen JL. Hydrographic changes in the eastern subpolar North Atlantic during the last deglaciation. *Quat Sci Rev* 2010, **29**(23): 3336-3345.

21. Johnson GC. Quantifying Antarctic Bottom Water and North Atlantic Deep Water volumes. *J Geophys Res-Oceans* 2008, **113**(C5): doi: 10.1029/2007jc004477.
22. Gebbie G. How much did Glacial North Atlantic Water shoal? *Paleoceanogr* 2014, **29**(3): 190-209.
23. Broecker W, Peacock SL, Weiss R, Fahrback E, Schroeder M, Mikolajewicz U, *et al.* How much deep water is formed in the Southern Ocean? *Journal of Geophysical Research* 1998, **103**(C8): 15833-15843.
24. Howe JNW, Piotrowski A, Noble TL, Mulitza S, Chiessi CM, Bayon G. North Atlantic Deep Water Production during the Last Glacial Maximum. *Nat Commun* 2016, **7**: doi: 10.1038/ncomms11765.
25. Toggweiler JR, Murnane R, Carson S, Gnanadesikan A, Sarmiento JL. Representation of the carbon cycle in box models and GCMs: 2. Organic pump. *Glob Biogeochem Cycle* 2003, **17**(1): doi:10.1029/2001gb001841.
26. Toggweiler JR, Gnanadesikan A, Carson S, Murnane R, Sarmiento JL. Representation of the carbon cycle in box models and GCMs: 1. Solubility pump. *Glob Biogeochem Cycle* 2003, **17**(1): doi:10.1029/2001gb001401.
27. Palter JB, Lozier MS, Barber RT. The effect of advection on the nutrient reservoir in the North Atlantic subtropical gyre. *Nature* 2005, **437**(7059): 687-692.

REVIEWERS' COMMENTS:

Reviewer #1 (Remarks to the Author):

Review of the revised version of 'More efficient North Atlantic carbon pump during the Last Glacial Maximum'

The authors have done a good job clarifying their correction method. The origin of the misunderstanding is in the use of the words 'carbon pumps'. What the authors refer to as 'part (ii)' of the solubility and biological carbon pumps, I would call the 'disequilibrium component' using the Ito & Follows (2005) language.

One issue is still unclear to me: are the authors suggesting that GNAIW in its formation region was supersaturated with respect to atmospheric CO₂? But in that case, there should have been a net outgassing of carbon from the ocean to the atmosphere in the subpolar North Atlantic during the LGM. So where did the ocean take up carbon then? In the Southern Ocean?

Reference

Ito, T., and Follows, M.J., J. Mar. Res. 63, 813-839 (2005)

Reviewer #2 (Remarks to the Author):

The authors have been very thorough in their revisions and responses to the comments that I raised in my review. I think this is an excellent, high quality manuscript, very suitable for publication in Nature Communications. I look forward to seeing it in print.

Reviewer #3 (Remarks to the Author):

I think the authors did a great job in addressing the comments raised by the reviewers. The authors emphasized the pragmatic recipe in the main text, which helped clarify the misunderstanding of the reviewers. They also added a section to quantify CO₂ uptake change between the Holocene and LGM following my suggestion, which is very critical to highlight the importance of North Atlantic in carbon pump during last glacial maximum. The calculation detail was also given in the Methods and it looks robust to me.

As I was generally satisfied with the initial draft and the authors made great progress following the suggestion of all three reviewers, I still recommend its publication in Nature communications and hope it will stimulate more discussions about the carbon pump in North Atlantic during glacial-interglacial cycles.

REVIEWERS' COMMENTS:

Reviewer #1 (Remarks to the Author):

Review of the revised version of 'More efficient North Atlantic carbon pump during the Last Glacial Maximum'

The authors have done a good job clarifying their correction method. The origin of the misunderstanding is in the use of the words 'carbon pumps'. What the authors refer to as 'part (ii)' of the solubility and biological carbon pumps, I would call the 'disequilibrium component' using the Ito & Follows (2005) language.

One issue is still unclear to me: are the authors suggesting that GNAIW in its formation region was supersaturated with respect to atmospheric CO₂? But in that case, there should have been a net outgassing of carbon from the ocean to the atmosphere in the subpolar North Atlantic during the LGM. So where did the ocean take up carbon then? In the Southern Ocean?

Reference

Ito, T., and Follows, M.J., J. Mar. Res. 63, 813-839 (2005)

We are pleased that the misunderstanding has been resolved.

We appreciate the Reviewer's point about "disequilibrium component" from Ito & Follows (2005). To avoid any confusion, in this study we stick with the air-sea exchange component CO₂ idea from Broecker and Peng (1992)¹. We will try to discuss various carbon cycle related parameters in future studies.

Previous proxy reconstructions^{2,3} suggest that the subpolar North Atlantic had lower surface *p*CO₂ than atmospheric *p*CO₂ during the LGM. This implies that surface waters in this region was not supersaturated and remained as a sink of atmospheric CO₂ during the LGM. We believe that the atmospheric CO₂ was sequestered in the glacial ocean by (i) increased CO₂ uptake efficiency in the North Atlantic (this study) and (ii) reduced CO₂ outgassing in the Southern Ocean. This has been discussed in the main text (see our discussion in last two paragraphs of the main text).

Finally, we thank the Reviewer for insightful comments that have significantly improved/clarified our manuscript.

Reviewer #2 (Remarks to the Author):

The authors have been **very thorough** in their revisions and responses to the comments that I raised in my review. I think this is **an excellent, high quality** manuscript, **very suitable** for publication in Nature Communications. I look forward to seeing it in print.

Thank you for your support to our study.

Reviewer #3 (Remarks to the Author):

I think the authors did a **great job** in addressing the comments raised by the reviewers. The authors emphasized the pragmatic recipe in the main text, which helped clarify the misunderstanding of the reviewers. They also added a section to quantify CO₂ uptake change between the Holocene and LGM following my suggestion, which is very critical to highlight the importance of North Atlantic in carbon pump during last glacial maximum. The calculation detail was also given in the Methods and it looks robust to me.

As I was generally satisfied with the initial draft and the authors **made great progress** following the suggestion of **all three reviewers**, I still recommend its publication in Nature communications and hope it will stimulate more discussions about the carbon pump in North Atlantic during glacial-interglacial cycles.

We are pleased to see that the Reviewer is satisfied with our responses. Thank you very much for your valuable advice, as well as for spending time to go through comments/responses from the other two reviewers.

References

1. Broecker W, Peng TH. Interhemispheric transport of carbon dioxide by ocean circulation. *Nature* 1992, **356**: 587-589.
2. Barker S, Elderfield H. Foraminiferal calcification response to glacial-interglacial changes in atmospheric CO₂. *Science* 2002, **297**(5582): 833-836.
3. Yu J, Thornalley DJR, Rae J, McCave IN. Calibration and application of B/Ca, Cd/Ca, and $\delta^{11}\text{B}$ in *Neogloboquadrina pachyderma* (sinistral) to constrain CO₂ uptake in the subpolar North Atlantic during the last deglaciation. *Paleoceanogr* 2013(28): 237-252.